# Sparsity-Agnostic Linear Bandits
# with Adaptive Adversaries

**Tianyuan Jin**
Department of Electrical and Computer Engineering
National University of Singapore
Singapore
tianyuan@nus.edu.sg

**Kyoungseok Jang**
Dipartimento di Informatica
Università degli Studi di Milano
Milano, Italy
ksajks@gmail.com

**Nicolò Cesa-Bianchi**
Università degli Studi di Milano
Politecnico di Milano
Milano, Italy
nicolo.cesa-bianchi@unimi.it

## Abstract

We study stochastic linear bandits where, in each round, the learner receives a set of actions (i.e., feature vectors), from which it chooses an element and obtains a stochastic reward. The expected reward is a fixed but unknown linear function of the chosen action. We study *sparse* regret bounds, that depend on the number $S$ of non-zero coefficients in the linear reward function. Previous works focused on the case where $S$ is known, or the action sets satisfy additional assumptions. In this work, we obtain the first sparse regret bounds that hold when $S$ is unknown and the action sets are adversarially generated. Our techniques combine online to confidence set conversions with a novel randomized model selection approach over a hierarchy of nested confidence sets. When $S$ is known, our analysis recovers state-of-the-art bounds for adversarial action sets. We also show that a variant of our approach, using Exp3 to dynamically select the confidence sets, can be used to improve the empirical performance of stochastic linear bandits while enjoying a regret bound with optimal dependence on the time horizon.

## 1 Introduction

$K$-armed bandits are a basic model of sequential decision-making in which a learner sequentially chooses which arm to pull in a set of $K$ arms. After each pull, the learner only observes the reward returned by the chosen arm. After $T$ pulls, the learner must obtain a total reward as close as possible to the reward obtained by always pulling the overall best arm. Linear bandits extend $K$-armed bandits to a setting in which arms belong to a $d$-dimensional feature space. In each round $t$ of a linear bandit problem, the learner receives an action set $\mathcal{A}_t \subset \mathbb{R}^d$ from the environment, chooses an arm $A_t \in \mathcal{A}_t$ based on the past observations, and then receives a reward $X_t$. In this work, we consider the stochastic setting in which rewards are defined by $X_t = \langle \theta_*, A_t \rangle + \varepsilon_t$, where $\theta_* \in \mathbb{R}^d$ is a fixed latent parameter and $\varepsilon_t$ is zero-mean independent noise. In linear bandits, the learner's goal is to minimize the difference between the total reward obtained by pulling in each round $t$ the arm $a \in \mathcal{A}_t$ maximizing $\langle \theta_*, a \rangle$ and the total reward obtained by the learner.

In stochastic linear bandits, the regret after $T$ rounds is known to be of order $d\sqrt{T}$ up to logarithmic factors. The linear dependence on the number $d$ of features implies that the learner is better off by ignoring features corresponding to negligible components of the latent target vector $\theta_*$. Hence,

38th Conference on Neural Information Processing Systems (NeurIPS 2024).

one would like to design algorithms that depend on the number $S \ll d$ of relevant features without requiring any preliminary knowledge on $\theta_*$. This is captured by the setting of sparse linear bandits, where $\theta_*$ is assumed to have only $0 < S \leq d$ nonzero components.

In the sparse setting, Lattimore and Szepesvári [17, Section 24.3] show that a regret of $\Omega\left(\sqrt{SdT}\right)$ is unavoidable for any algorithm, even with knowledge of $S$. When $S$ is known, this lower bound is matched (up to log factors) by an algorithm of Abbasi-Yadkori et al. [2] who, under the same assumptions and for the same algorithm, also prove an instance-dependent regret bound of $\widetilde{O}\left(\frac{Sd}{\Delta}\right)$. Here $\Delta$ is the minimum gap, over all $T$ rounds, between the expected reward of the optimal arm and that of any suboptimal arm. In this work we focus on the sparsity-agnostic setting, i.e., when $S$ is unknown. Fewer results are known for this case, and all of them rely on additional assumptions on the action set, or assumptions on the sparsity structure. For example, if the action set is stochastically generated, Oh et al. [22] prove a $\widetilde{O}\left(S\sqrt{T}\right)$ sparsity-agnostic regret bound. More recently, Dai et al. [9] showed a sparsity-agnostic bound $\widetilde{O}\left(S^2\sqrt{T} + S\sqrt{dT}\right)$ when the action set is fixed and equal to the unit sphere. In a model selection setting, Cutkosky et al. [8] prove a $\widetilde{O}\left(S^2\sqrt{T}\right)$ sparsity-agnostic regret bound for adversarial action sets, but under an additional nestedness assumption: $(\theta_*)_i \neq 0$ for $i = 1, \ldots, S$. Surprisingly, no bounds improving on the $\widetilde{O}\left(d\sqrt{T}\right)$ regret of the OFUL algorithm [1] in the sparsity-agnostic case are known that avoid additional assumptions on the sparsity structure or on the action set generation.

**Main contributions.** Here is the summary of our main contributions. All the proofs of our results can be found in the appendix.

• We introduce a randomized sparsity-agnostic linear bandit algorithm, SparseLinUCB, achieving regret $\widetilde{O}\left(S\sqrt{dT}\right)$ with no assumptions on the sparsity structure (e.g., nestedness) or on the action set (which may be controlled by an adaptive adversary). When $S = o(\sqrt{d})$, our bound is strictly better than the OFUL bound $\widetilde{O}\left(d\sqrt{T}\right)$.

• Our analysis of SparseLinUCB simultaneously guarantees an instance-dependent regret bound $\widetilde{O}\left(\max\{d^2, S^2 d\}/\Delta\right)$, where $\Delta$ is the smallest suboptimality gap over the $T$ rounds.

• If the sparsity level is known, our algorithm recovers the optimal bound $\widetilde{\Theta}(\sqrt{SdT})$.

• We also introduce AdaLinUCB, a variant of SparseLinUCB that uses Exp3 to learn the probability distribution over a hierarchy of confidence sets in stochastic linear bandits. Unlike previous works, which only showed a $\widetilde{O}\left(T^{2/3}\right)$ regret bound for similar approaches, AdaLinUCB has a $\widetilde{O}\left(\sqrt{T}\right)$ regret bound. In experiments on synthetic data, AdaLinUCB performs better than OFUL.

**Technical challenges.** Recall that the arm chosen in each round by OFUL is

$$A_t = \operatorname*{argmax}_{a \in \mathcal{A}_t} \langle a, \widehat{\theta}_t \rangle + \sqrt{\gamma_t} \|a\|_{V_{t-1}^{-1}} \tag{1.1}$$

where $\widehat{\theta}_t$ is the regularized least-squares estimate of $\theta_*$, $V_{t-1} = I + \sum_{s<t} A_s A_s^\top$ is the regularized covariance matrix of past actions, and $\sqrt{\gamma_t}$ is the radius of the confidence set

$$\left\{ \theta \in \mathbb{R}^d : \|\theta - \widehat{\theta}_{t-1}\|_{V_{t-1}}^2 \leq \gamma_t \right\} . \tag{1.2}$$

The squared radius $\gamma_t = O(d \ln t)$ is such that $\theta_*$ belongs to (1.2) with high probability simultaneously for all $t \geq 1$. Our approach, instead, builds on the online to confidence set conversion technique of Abbasi-Yadkori et al. [2], where they show how to design a different confidence set for $\theta_*$ based on the predictions of an arbitrary algorithm for online linear regression, such that the squared radius of the confidence set is roughly equal to the regret bound of the algorithm. Using the algorithm SeqSEW for sparse online linear regression [13], whose regret bound is $O(S \log T)$, they obtain the optimal regret $\widetilde{O}\left(\sqrt{SdT}\right)$ for sparse linear bandits. Unfortunately, this result requires knowing $S$ to properly set the radius of the confidence set. Our strategy SparseLinUCB (Algorithm 1) bypasses this problem by running the online to confidence set conversion technique over a hierarchy of nested confidence sets with radii $\alpha_i = 2^i \log T$ for $i = 1, \ldots, n = \Theta(\log d)$. The framework of Abbasi-Yadkori et al. [2] guarantees that, for any sparsity value $S \in [d]$, there is a critical radius $\alpha_o = O(S \log T)$ such that, with high probability, $\theta_*$ lies in the set with radius $\alpha_i$ for all $i \geq o$. SparseLinUCB randomizes the choice of the index $i$ of the confidence radius $\alpha_i$, used for selecting the action at time $t$. If

Table 1: Comparison with other sparse linear bandit works. $S \in [d]$ is the sparsity level and $\Delta$ is the suboptimality gap (3.3). The nested assumption refers to $(\theta_*)_i \neq 0$ for $i = 1, \ldots, S$. The minimum signal and the compatibility condition refer to assumptions on the distribution of the action set and on the smallest value of the non-zero elements in $\theta_*$. Smoothed adversary refers to adversarially selected action sets with added Gaussian noise. The regret bounds listed in [1, 2, 23, 8, 25, 18, 9] are high-probability bounds: with high probability, the regret is of the same order as the bound in the table.

| Reference | Sparsity Agnostic | Adaptive Adversary | Expected Regret | Assumptions |
|---|---|---|---|---|
| Abbasi-Yadkori et al. [1] | ✓ | ✓ | $\widetilde{O}(\min\{d\sqrt{T}, d^2/\Delta\})$ | - |
| Abbasi-Yadkori et al. [2] | ✗ | ✓ | $\widetilde{O}(\min\{\sqrt{SdT}, dS/\Delta\})$ | - |
| Pacchiano et al. [23, 24] | ✓ | ✗ | $\widetilde{O}(S^2\sqrt{T})$ | Nested, i.i.d. actions |
| Cutkosky et al. [8] | ✓ | ✓ | $\widetilde{O}(S^2\sqrt{T})$ | Nested |
| Pacchiano et al. [25] | ✓ | ✓ | $\widetilde{O}(S^2\sqrt{T})$ | Nested |
| | ✓ | ✗ | $\widetilde{O}(S^2 d^2/\Delta)$ | Nested, i.i.d. actions |
| Lattimore et al. [18] | ✗ | ✗ | $\widetilde{O}(S\sqrt{T})$ | Action set is hypercube $\varepsilon_t \in [-1, 1]$ |
| Sivakumar et al. [26] | ✗ | ✓ | $\widetilde{O}(S\sqrt{T})$ | Smoothed adversary |
| Hao et al. [15] | ✗ | ✗ | $\widetilde{O}(\sqrt{ST})$ | Actions set spans $\mathbb{R}^d$ Minimum signal |
| Oh et al. [22] | ✓ | ✗ | $\widetilde{O}(S\sqrt{T})$ | Compatibility |
| Dai et al. [9] | ✓ | ✗ | $\widetilde{O}(S^2\sqrt{T} + S\sqrt{dT})$ | Action set is unit sphere |
| Lower bound [17] | ✗ | ✓ | $\Omega(\sqrt{SdT})$ | - |
| **This paper** | ✓ | ✓ | $\widetilde{O}\left(\min\{S\sqrt{dT}, \frac{1}{\Delta}\max\{d^2, S^2d\}\}\right)$ | - |
| **This paper** | ✗ | ✓ | $\widetilde{O}(\sqrt{SdT})$ | - |

the random index $I_t$ is such that $I_t \geq o$, then we can bound the regret incurred at step $t$ using standard techniques [1, 2]. By choosing $\mathbb{P}(I_t = i)$ proportional to $2^{-i}$, we make sure that larger confidence sets (delivering suboptimal regret bounds) are chosen with exponentially small probability. If $I_t < o$, then $\theta_*$ is not guaranteed to lie in the confidence set of radius $\alpha_{I_t}$ with high probability. Our main technical contribution is to show that the regret summed over these bad rounds is bounded by $\widetilde{O}\left(\sqrt{SdT/Q}\right)$, where $Q = \mathbb{P}(I_t \geq o)$. The proof of this bound requires showing that the regret in a bad round $t$ (when $I_t < o$) can be bounded by $\sqrt{\alpha_o}\|A_t^o\|_{V_{t-1}^{-1}}$. Proving that $\|A_t^o\|_{V_{t-1}^{-1}}$ shrinks fast enough uses the fact that $\det V_t$ grows fast enough due to the exploration in the good rounds $t$ (when $I_t \geq o$). This is done by a carefully designed peeling technique that partitions $[T]$ in blocks based on the value of $\det V_t$.

To extend our analysis of `SparseLinUCB` and obtain instance-dependent regret bound, we apply the techniques of Abbasi-Yadkori et al. [1] to show that the regret over the good rounds is bounded by $\widetilde{O}(d^2/\Delta)$. The regret over a bad round $t$ is controlled by $(\alpha_o/\Delta)\|A_t^o\|_{V_{t-1}^{-1}}^2$ and—using techniques similar to the instance-independent analysis—we bound the regret summed over all bad rounds with $\widetilde{O}(Sd/(Q\Delta))$.

Given that `SparseLinUCB` uses a fixed probability of order $2^{-i}$ to choose its confidence radius $\alpha_i$, it is tempting to explore adaptive probability assignments, that increase the probability of a confidence set proportionally to the rewards obtained by the actions that were selected based on that set. Algorithm `AdaLinUCB` (see Algorithm 2) is a variant of `SparseLinUCB` using Exp3 [4] to assign probabilities to confidence sets. The analysis of `AdaLinUCB` combines—in a non-trivial way—the analysis of Exp3 (including a forced exploration term $q$) with that of `SparseLinUCB`. Although the resulting regret bound does not improve on `OFUL`, our algorithm provides a new principled solution to the problem of tuning the radius in (1.1). Experiments show that `SparseLinUCB` can perform better than `OFUL`.

## 1.1 Additional related work

**Sparse linear bandits.** With the goal of obtaining sparsity-agnostic regret bounds, different types of assumptions on the action set have been considered in the past. Starting from the $\widetilde{O}(S\sqrt{T})$ regret upper bound of [18], where the action set is assumed fixed and equal to the hypercube, some works considered stochastic action sets and proved regret bounds depending on spectral parameters of the action distribution, such as the minimum eigenvalue of the covariance matrix [9, 15, 16, 20]. Others assumed a stochastic action set with strong properties, such as compatibility conditions or margin conditions [3, 6, 7, 19, 22]. As far as we know, there has been no research on adaptive adversarial action sets after [2].

**Model selection.** Sparse linear bandits can be naturally viewed as a bandit model selection problem. For example, Ghosh et al. [14] establish a regret bound of $\widetilde{O}(\sqrt{ST} + d^2/\alpha^{4.65})$ for a fixed action set, where $\alpha$ is the minimum absolute value of the nonzero components of $\theta_*$. Quite a bit of work has been devoted to sparse regret bounds in the nested setting. With i.i.d. and fixed-size actions sets, Foster et al. [12] achieve a regret bound of order $\widetilde{O}(S^{1/3}T^{2/3}/\gamma^3)$ in the nested setting, where $\gamma$ is the smallest eigenvalue of the covariance matrix of $\mathcal{A}_t$. Under the same assumption on the action set, Pacchiano et al. [24, 23] obtain a regret bound of $\widetilde{O}(S^2\sqrt{T})$. For adversarial action sets, Cutkosky et al. [8] obtain $\widetilde{O}(S^2\sqrt{T})$ in the nested setting. When actions are sampled i.i.d., Cutkosky et al. [8] and Pacchiano et al. [25] obtain a regret bound of $\widetilde{O}(S^2\sqrt{T})$ for nested settings. They also obtain simultaneous instance-dependent bounds, in particular, Pacchiano et al. [25] achieve $\widetilde{O}((Sd)^2/\Delta)$. Compared to the instance-dependent regret bound, our results are more general, as we allow the action set to be adaptively chosen by an adversary and do not require the nested assumption.

**Parameter tuning.** Although the theoretical anlysis of `OFUL` only holds for $\gamma_t = O(d \log t)$, smaller choices of the radius in (1.2) are known to perform better in practice. Our design of `SparseLinUCB` and `AdaLinUCB` borrows ideas from the parameter tuning setting, which is typically addressed using a set of base algorithms and a randomized master algorithm that adaptively changes the probability of selecting each base algorithm [21, 23, 24]. In particular, `AdaLinUCB` builds on [11], where they show that running `Exp3` as the master algorithm over instances of `OFUL` with different radii has a better empirical performance than Thompson Sampling and UCB. Yet, they only show a regret bound of $\widetilde{O}(T^{2/3})$ when the action set is drawn i.i.d. in each round (they also prove a bound of order $\sqrt{T}$, but only under additional assumptions on the best model). This is consistent with the results of Pacchiano et al. [24], who also obtained a regret of the same order using `Exp3` as master algorithm.

## 2 Problem definition

In linear bandits, a learner and an adversary interact over $T$ rounds. In each round $t = 1, \ldots, T$:

1. The adversary chooses an arm set $\mathcal{A}_t \subset \mathbb{R}^d$;
2. The learner choose an arm $A_t \in \mathcal{A}_t$;
3. The learner obtains a reward $X_t$.

We assume the adversary is adaptive, i.e., $\mathcal{A}_t$ can depend in an arbitrary way on the (possibly randomized) past choices of the learner. The reward in each round $t$ satisfies

$$X_t = \langle A_t, \theta_* \rangle + \varepsilon_t. \tag{2.1}$$

Here $\theta_* \in \mathbb{R}^d$ is a fixed and unknown target vector and $\{\varepsilon_t\}_{t \in [T]}$ are independent conditionally 1-subgaussian random variables. We also assume $\|\theta_*\|_2 \leq 1$ and $\|a\|_2 \leq 1$ for all $a \in \mathcal{A}_t$ and all $t \in [T]$.[1] The regret of a strategy over $T$ rounds is defined as the difference between the reward obtained by the optimal policy, always choosing the best arm in $\mathcal{A}_t$, and the reward obtained by the strategy choosing arm $A_t \in \mathcal{A}_t$ for $t \in [T]$,

$$R_T = \sum_{t=1}^{T} \max_{a \in \mathcal{A}_t} \langle a, \theta_* \rangle - \sum_{t=1}^{T} \langle A_t, \theta_* \rangle.$$

---

[1]The choice of the constant 1 is arbitrary. Choosing different constants would scale our bounds similarly to the scaling of the bounds in [2].

In the sparse setting, we would like to devise strategies whose regret depends on

$$S = \|\theta_*\|_0 = \sum_{i=1}^{d} \mathbb{1}\{\theta_i \neq 0\}$$

corresponding to the number of nonzero components of $\theta_*$.

## 2.1 Online to confidence set conversions

Establishing a confidence set including the target vector with high probability is at the core of linear bandit algorithms, and our approach for designing a sparsity-agnostic algorithm is based on a result by Abbasi-Yadkori et al. [2]. They show how to construct a confidence set for OFUL [1] based on the predictions of a generic algorithm for online linear regression, a sequential decision-making setting defined as follows. For $t = 1, \ldots, T$:

1. The adversary privately chooses input $A_t \in \mathbb{R}^d$ and outcome $X_t \in \mathbb{R}$;

2. The learner observes $A_t$ and chooses prediction $\widehat{X}_t \in \mathbb{R}$;

3. The adversary reveals $X_t$ and the learner suffers loss $(\widehat{X}_t - X_t)^2$.

The learner's goal in online linear regression is to minimize the following notion of regret against any comparator $\theta \in \mathbb{R}^d$

$$\rho_T(\theta) = \sum_{t=1}^{T}(X_t - \widehat{X}_t)^2 - \sum_{t=1}^{T}(X_t - \langle A_t, \theta\rangle)^2.$$

The confidence set proposed by Abbasi-Yadkori et al. [2] is established by the following result.

**Lemma 2.1** (Abbasi-Yadkori et al. [2, Corollary 2]). *Let $\delta \in (0, 1/4]$ and $\|\theta_*\|_2 \leq 1$. Assume a sequence $\{(A_t, X_t)\}_{t\in[T]}$, where $X_t$ satisfies (2.1) for all $t \in [T]$, is fed to an online linear regression algorithm $\mathcal{B}$ generating predictions $\{\widehat{X}_t\}_{t\in[T]}$. Then $\mathbb{P}\big(\exists t \in [T] : \theta_* \notin \mathcal{C}_t\big) \leq \delta$, where*

$$\mathcal{C}_t = \left\{\theta \in \mathbb{R}^d : \|\theta\|_2^2 + \sum_{s=1}^{t-1}(\widehat{X}_s - \langle A_s, \theta\rangle)^2 \leq \gamma(\delta, \theta_*)\right\} \tag{2.2}$$

$$\gamma(\delta, \theta_*) = 2 + 2B_T(\theta_*) + 32\log\left(\frac{\sqrt{8} + \sqrt{1 + B_T(\theta_*)}}{\delta}\right)$$

*and $B_T(\theta_*)$ is an upper bound on the regret $\rho_T(\theta_*)$ of $\mathcal{B}$. When understood from the context, we will abuse the notation and denote the best radius in hindsight by $\gamma(\delta) := \gamma(\delta, \theta_*)$ and the regret bound by $B_T$.*

Gerchinovitz [13] designed an algorithm, SeqSEW, for *sparse* linear regression that bounds $\rho_T(\theta)$ in terms of $\|\theta\|_0$ simultaneously for all comparators $\theta \in \mathbb{R}^d$. Below here, we state his bound in the formulation of Lattimore and Szepesvári [17].

**Lemma 2.2** (Lattimore and Szepesvári [17, Theorem 23.6]). *Assume $\max_{t\in[T]} \|A_t\|_2 \leq 1$ and $\max_{t\in[T]} |X_t| \leq 1$. There exists a universal constant $c$ such that algorithm SeqSEW achieves, for any $\theta \in \mathbb{R}^d$,*

$$\rho_T(\theta) \leq B_T(\theta) := c\|\theta\|_0 \left\{\log(e + T^{1/2}) + C_T \log\left(1 + \frac{\|\theta\|_1}{\|\theta\|_0}\right)\right\}$$

*where $C_T = 2 + \log_2 \log(e + T^{1/2})$.*

Using the confidence set (2.2) with $\mathcal{B}$ set to SeqSEW, Abbasi-Yadkori et al. [2] achieved the minimax optimal regret bound of $\widetilde{O}(\sqrt{SdT})$. However, to construct $\mathcal{C}_t$, the learner must know $\gamma(\delta)$, which depends on the unknown sparsity level $S = \|\theta_*\|_0$ through $B_T$.

---

**Algorithm 1** SparseLinUCB

---

1: **Input:** $T \in \mathbb{N}$, $n \in \mathbb{N}$ and $\mathbf{q} \in \Delta_n := \{(q_1, \ldots, q_n) \in [0,1]^n : \sum_{i=1}^n q_i = 1\}$
2: **Initialization:** Let $V_0 = I$, $\widehat{\theta}_0 = (0, \ldots, 0)$
3: **for** $t = 1, 2, \ldots, T$ **do**
4:    Receive action set $\mathcal{A}_t$ and draw $I_t$ from distribution $\mathbf{q}$
5:    Choose action $A_t = \underset{a \in \mathcal{A}_t}{\operatorname{argmax}} \left( \langle a, \widehat{\theta}_{t-1} \rangle + \|a\|_{V_{t-1}^{-1}} \sqrt{2^{I_t} \log T} \right)$
6:    Receive reward $X_t$
7:    $V_t = V_{t-1} + A_t A_t^\top$
8:    Feed $(A_t, X_t)$ to SeqSEW and obtain prediction $\widehat{X}_t$
9:    Compute regularized least squares estimate $\widehat{\theta}_t = \underset{\theta \in \mathbb{R}^d}{\operatorname{argmin}} \left( \|\theta\|_2^2 + \sum_{s=1}^t \left( \widehat{X}_s - \langle \theta, A_s \rangle \right)^2 \right)$
10: **end for**

---

## 3 A multi-level sparse linear bandit algorithm

In this section, we introduce our main algorithm, SparseLinUCB, whose pseudo-code is shown in Algorithm 1. The algorithm, which runs SeqSEW as base algorithm $\mathcal{B}$, uses a hierarchy of confidence sets of increasing radius. In each round $t = 1, \ldots, T$, after receiving the action set $\mathcal{A}_t$, the algorithm draws the index $I_t$ of the confidence set for time $t$ by sampling from the distribution $\mathbf{q} \in \Delta_n := \{(q_1, \ldots, q_n) \in [0,1]^n : \sum_{i=1}^n q_i = 1\}$. Then the algorithm plays the action $A_t$ using the confidence set $\mathcal{C}_t^{I_t} := \{\theta \in \mathbb{R}^d : \|\theta - \widehat{\theta}_{t-1}\|_{V_{t-1}}^2 \le 2^{I_t} \log T\}$ (where a larger $I_t$ implies a larger radius, and thus more exploration). Following the online to confidence set approach, upon receiving the reward $X_t$, the algorithm feeds the pair $(A_t, X_t)$ to SeqSEW and uses the prediction $\widehat{X}_t$ to update the regularized least squares estimate $\widehat{\theta}_t$.

Let $\alpha_i = 2^i \log T$ for all $i \in \mathbb{N}$ and set $n \in \mathbb{N}$ as

$$n = \left\lceil \log_2 \frac{\underset{\theta \in \mathbb{R}^d : \|\theta\|_2 \le 1}{\max} \gamma(1/T, \theta)}{\log T} \right\rceil \tag{3.1}$$

where $\gamma(\delta, \theta)$ is defined in Lemma 2.1 for $\mathcal{B} = $ SeqSEW.

One can check that $n = \Theta(\log d)$ (when $\|\theta\|_0 = d$), which gives $\alpha_n = \Theta(d \log T)$. Our bounds depend on the following quantity, which defines the index of the smallest "safe" confidence set (i.e., the smallest $i \in [n]$ such that $\theta_* \in \mathcal{C}_t^i$ for all $t \in [T]$),

$$o := \underset{i \in [n]}{\operatorname{argmin}} \left\{ \gamma(1/T) \le \alpha_i \right\} \tag{3.2}$$

The choice of our confidence set (Line 4 in Algorithm 1) is justified by the following result, which implies that $o$ is safe.

**Lemma 3.1.** *For $\mathcal{C}_t$ defined in (2.2), we have that $\mathcal{C}_t \subseteq \mathcal{C}_t^o$ for all $t \in [T]$.*

As we use SeqSEW as base algorithm $\mathcal{B}$, $\alpha_o = O(S \log T)$. Our main result is an upper bound on the regret of SparseLinUCB.

**Theorem 3.2.** *The expected regret of SparseLinUCB run with the number of models $n$ in (3.1) and a distribution $\mathbf{q} = \{q_s\}_{s \in [n]}$ satisfies*

$$\mathbb{E}[R_T] = O\left( (\log T) \sum_{s \ge o} \sqrt{d 2^s T q_s} + (\log T) \sqrt{STd/Q} \right)$$

*where $Q = \sum_{s \ge o} q_s$.*

If the sparsity level $S$ is indeed known, then $o$ in (3.2) can be computed and we get the following bound, which is tight up to log factors [17].

**Corollary 3.3.** *Assume that the sparsity level $S$ is known and choose the number of models $n > o$ and the distribution $\{q_s\}_{s \in [n]}$ with $q_o = 1$, where $o$ is set as in (3.2). Then, the expected regret of* `SparseLinUCB` *is $\mathbb{E}[R_T] = O\big(\sqrt{SdT} \log T\big)$.*

**An instance-dependent bound.** `SparseLinUCB` also enjoys an instance-dependent regret bound comparable to that of `OFUL`. Let $\Delta$ be the minimum gap between the optimal arm and any suboptimal arms over all rounds,

$$\Delta = \min_{t \in [T]} \min_{a \in \mathcal{A}_t \setminus A_t^*} \langle A_t^* - a, \theta_* \rangle, \tag{3.3}$$

where $A_t^* = \max_{a \in \mathcal{A}_t} \langle a, \theta_* \rangle$ is the optimal arm for round $t$.

**Theorem 3.4.** *The expected regret of* `SparseLinUCB` *run with the number of models $n$ in (3.1), a distribution $\boldsymbol{q} = \{q_s\}_{s \in [n]}$ and using* `SeqSEW` *as base algorithm satisfies*

$$\mathbb{E}[R_T] = O\left( \frac{(dS/Q) + d^2}{\Delta} (\log T)^2 \right)$$

*where $Q = \sum_{s \geq o} q_s$.*

**Sparsity-agnostic tuning of randomization.** Next, we look at a specific choice of **q**. Fix $C \geq 1$ and let

$$q_s = \begin{cases} C^2 2^{-s} & \text{if } C^2 2^{-s} < 1 \\ \kappa & \text{otherwise,} \end{cases} \tag{3.4}$$

where $\kappa > 0$ is chosen so to normalize the probabilities. It is easy to verify that for any $C \geq 1$,

$$\sum_{s \in [n]} \mathbb{1}\big\{C^2 2^{-s} < 1\big\} q_s \leq 1$$

implying that $\kappa$ can be chosen in $[0, 1]$. Combining Theorem 3.2 and 3.4, we obtain the following corollary providing a hybrid distribution-free and distribution-dependent bound.

**Corollary 3.5.** *Pick any $C \geq 1$. Let the number of models $n$ as in (3.1) and $\boldsymbol{q} = \{q_s\}_{s \in [n]}$ be chosen as in (3.4). Then the expected regret of* `SparseLinUCB` *is*

$$\mathbb{E}[R_T] = \widetilde{O}\left( \min \left\{ \max \big\{C, S/C\big\} \sqrt{dT}, \frac{\max\big\{d^2, S^2 d/C^2\big\}}{\Delta} \right\} \right)$$

For $C = 1$ the above bound is $\widetilde{O}(S\sqrt{dT})$, which is tight up to the factor $\sqrt{S}$ due to the lower bound of $\Omega(\sqrt{SdT})$ [17]. However, as mentioned in Lattimore and Szepesvári [17, Section 23.5], no algorithm can enjoy the regret of $\widetilde{O}(\sqrt{SdT})$ simultaneously for all possible sparsity levels $S$. While our worst-case regret bound improves with a smaller $S$, the problem-dependent regret bound scales at least as $(d^2/\Delta) \log T$, which is independent of $S$. This raises an interesting question: could the problem-dependent bound also benefit from sparsity? Even with a very small probability $p$ of choosing radius $\alpha_n$, the expected number of steps using $\alpha_n$ would be $pT$. The results in [2] demonstrate that running the OFUL algorithm with $\alpha_n$ over $pT$ steps results in a regret of $\widetilde{O}(d^2/\Delta)$. One simple way is to decrease the frequency of selecting radius $\alpha_n$. However, selecting $\alpha_n$ less than $d^2/\Delta^2$ times may prevent the algorithm from obtaining a good enough estimate of $\theta_*$ in certain settings.

**Remark 3.6.** *At first glance, it may seem straightforward to select $C$ in Corollary 3.5, as setting $C = \sqrt{d}$ yields a regret of $\widetilde{O}(d^2/\Delta)$ without apparent trade-offs. However, the trade-off lies in balancing the instance-dependent and worst-case regret bounds. Opting for $C = d$ indeed yields an instance-dependent bound of $\widetilde{O}(d^2/\Delta)$. However, this comes at the expense of the worst-case bound, which remains $\widetilde{O}(d\sqrt{T})$, negating any advantages derived from the sparsity assumption $S \ll d$.*

If the sparsity level $S$ is indeed known, then $o$ in (3.2) can be computed and we get the following bound.

**Corollary 3.7.** *Assume that the sparsity level $S$ is known and choose $\{q_s\}_{s \in [n]}$ with $q_o = 1$, where $o$ is set as in (3.2). Then, the expected regret of* `SparseLinUCB` *is $\mathbb{E}[R_T] = \widetilde{O}\big(\frac{Sd}{\Delta}\big)$.*

We note that by setting $q_o = 1$ in Theorem 3.4, the regret bound becomes $\widetilde{O}(d^2/\Delta)$. This result, as detailed in Theorem 3.4, arises from the parameter $q_n > 0$. However, in this case, $q_n = 0$, which allows us to achieve a more favorable regret bound.

---

**Algorithm 2** `AdaLinUCB`

---

1: **Input:** $T \in \mathbb{N}$, $\eta > 0$, $q \in (0, 1]$

2: **Initialization:** Let $S_{i,0} = 0$ for all $i \in [n]$, $V_0 = I$, $\widehat{\theta}_0 = (0, \ldots, 0)$

3: **for** $t = 1, 2, \cdots, T$ **do**

4:     Receive action set $\mathcal{A}_t$ and draw a Bernoulli random variable $Z_t$ with $\mathbb{P}(Z_t = 1) = q$

5:     **if** $Z_t = 1$ **then**

6:         Choose optimistic action $A_t = \underset{a \in \mathcal{A}_t}{\operatorname{argmax}} \left( \langle a, \widehat{\theta}_{t-1} \rangle + \|a\|_{V_{t-1}^{-1}} \sqrt{2^n \log T} \right)$

7:         Receive reward $X_t$;

8:     **else**

9:         Draw $I_t$ from the distribution $P_{t,i} = \dfrac{\exp\left(\eta S_{t-1,i}\right)}{\sum_{j=1}^n \exp\left(\eta S_{t-1,j}\right)}$ for $i \in [n]$;

10:        Choose action $A_t = \underset{a \in \mathcal{A}_t}{\operatorname{argmax}} \left( \langle a, \widehat{\theta}_{t-1} \rangle + \|a\|_{V_{t-1}^{-1}} \sqrt{2^{I_t} \log T} \right)$

11:        Receive reward $X_t$;

12:        Compute $S_{t,j} = S_{t-1,j} - \dfrac{\mathbb{1}\{I_t = j\}(2 - X_t)/4}{P_{t,j}}$ for $j \in [n]$;

13:     **end if**

14:     $V_t = V_{t-1} + A_t A_t^\top$;

15:     Feed $(A_t, X_t)$ to `SeqSEW` and obtain prediction $\widehat{X}_t$;

16:     Compute regularized least squares estimate $\widehat{\theta}_t = \underset{\theta \in \mathbb{R}^d}{\operatorname{argmin}} \left( \|\theta\|_2^2 + \sum_{s=1}^t \left( \widehat{X}_s - \langle \theta, A_s \rangle \right)^2 \right)$;

17: **end for**

---

## 4 Adaptive model selection for stochastic linear bandits

`SparseLinUCB` is also designed to handle adaptive adversarial action sets. A crucial parameter of `SparseLinUCB` is $\{q_s\}_{s \in [n]}$, the distribution from which the radius of the confidence set is drawn. It is a natural question whether there exists an algorithm that adaptively updates this distribution based on the observed rewards. In this section we introduce `AdaLinUCB` (Algorithm 2), which runs `Exp3` to dynamically adjust the distribution used by `SparseLinUCB`.

`AdaLinUCB` takes as input a forced exploration term $q$ and the learning rate $\eta$ for `Exp3`. Similarly to `SparseLinUCB`, `AdaLinUCB` designs confidence sets of various radii, but its selection method differs in two aspects. First, with probability $q$, the algorithm performs exploration based on the confidence set with the largest radius. With probability $1 - q$, the algorithm instead draws the action based on `Exp3`. The distribution $P_t$ used by `Exp3` at round $t$ is based on exponential weights applied to the total estimated loss, denoted by $S_t$ (for technical reasons, we translate losses into rewards). The algorithm then draws $I_i$ from $P_t$ and selects the action $A_t$ based on the confidence set with radius $2^{I_t} \log T$. Finally, reward $X_t$ is observed and the pair $(A_t, X_t)$ is fed to `SeqSEW`. The prediction $\widehat{X}_t$ returned by `SeqSEW` is used to update the regularized least squares estimate $\widehat{\theta}_t$.

The following theorem states the theoretical regret upper bound of `AdaLinUCB`.

**Theorem 4.1.** *If the random independent noise $\varepsilon_t$ in (2.1) satisfies $\varepsilon_t \in [-1, 1]$ for all $t \in [T]$, then the regret of `AdaLinUCB` run with $\eta = \sqrt{(\log n)/(Tn)}$ for $n$ in (3.1) and $q \in (0, 1]$ satisfies*

$$\mathbb{E}[R_T] \leq \left( \sqrt{8\alpha_n q} + 4\sqrt{(2\alpha_o)/q} \right) \sqrt{dT \log\left( 1 + \frac{TL^2}{d} \right) + 1} + O\left( \sqrt{nT \log n} \right)$$

$$= \widetilde{O}\left( \max\left\{ \sqrt{dq}, \sqrt{S/q} \right\} \sqrt{dT} \right) .$$

Although `AdaLinUCB` dynamically adjusts the distribution used by `SparseLinUCB` and may achieve better empirical performance, its regret bound is no better than that of `SparseLinUCB`. The issue is that the action chosen by `AdaLinUCB` in Line 10 does not ensure enough exploration to control the regret. Consequently, the algorithm needs to choose the optimistic action in Line 6 with constant probability $q$. `SparseLinUCB` has a similar parameter, $Q$, that bounds from the above the probability

of choosing the optimistic action. The key difference is that $Q$ can be optimized for an unknown $S$ by carefully selecting the distribution $\mathbf{q} = \{q_s\}_{s \geq 1}$, whereas the parameter $q$ does not provide a similar flexibility.

## 5 Model selection experiments

In this section we describe some experiments we performed on synthetic data to verify whether `AdaLinUCB` could outperform `OFUL` in a model selection task. We also test the empirical performance of `SparseLinUCB` on the same data (additional details on all the algorithms and the experimental setting are in Appendix E). The data for our model selection experiments are generated using targets $\theta_*$ with different sparsity levels, as we know that sparsity affects the radius of the optimal confidence set. On the other hand, since no efficient implementation of `SeqSEW` is known [17, Section 23.5], we cannot implement the online to confidence set approach as described in [2] to capture sparsity. Instead, we run `SparseLinUCB` and `AdaLinUCB` with $\widehat{X}_t = X_t$ for all $t \in [T]$, which—due to the form of our confidence sets—amounts to running the algorithms over multiple instances of `OFUL` with different choices of radius $\alpha_i$ for $i \in [n]$.

We run `SparseLinUCB` and `AdaLinUCB` with $\alpha_i = \alpha_{i,t} = 2^i \log t$ (a mildly time-dependent choice) for $i = 0, 1, \cdots, \log_2 d$. We also include $\alpha_0 = 0$ corresponding to the greedy strategy $A_t = \arg\max_{a \in \mathcal{A}_t} \langle a, \widehat{\theta}_{t-1} \rangle$. The suffix `_Unif` indicates $\{q_s\}_{s \in [n]}$ set to $(\frac{1}{n}, \cdots, \frac{1}{n})$. The suffix `_Theory` indicates $q_s = \Theta(2^{-s})$ for $s = 0, \ldots, n$ as prescribed by (3.4). Finally, we included `SparseLinUCB_Known` using $q_i = \mathbb{1}\{i = o\}$ to test the performance when the optimal index $o$ (for the given $S$) is known in advance (see Corollary 3.3). We run our experiments with a fixed set of random actions, $\mathcal{A}_t = \mathcal{A}$ for all $t \in [T]$, where $|\mathcal{A}| = 30$ and $\mathcal{A}$ is a set of vectors drawn i.i.d. from the unit sphere in $\mathbb{R}^{16}$. The target vector $\theta_*$ is a $S$-sparse ($S = 1, 2, 4, 8, 16$) vector whose non-zero coordinates are drawn from the unit sphere in $\mathbb{R}^S$. The noise $\varepsilon_t$ is drawn i.i.d. from the uniform distribution over $[-1, 1]$. Each curve is an average over 20 repetitions with $T = 10^4$ where, in each repetition, we draw fresh instances of $\mathcal{A}$ and $\theta_*$.

As our implementations are not sparsity-aware, we cannot expect the regret to strongly depend on the sparsity level. Indeed, only the regret of `SparseLinUCB_Known` (which is tuned to the sparsity $S$) is significantly affected by sparsity. The theory-driven choice of $\{q_s\}_{s \in [n]}$ (`SparseLinUCB_Theory`) performs better than the uniform assignment (`SparseLinUCB_Unif`), and is in the same ballpark as `OFUL`. On the other hand, `AdaLinUCB_Unif` and `AdaLinUCB_Theory` outperform all the competitors, including `OFUL`. This provides evidence that using `Exp3` for adaptive model selection may significantly boost the empirical performance of stochastic linear bandits.

## 6 Limitations and open problems

Unlike previous works, we prove sparsity-agnostic regret bounds with no assumptions on the action sets or on $\theta_*$ (other than boundedness of $\|\theta_*\|$ and $\|a\|$ for $a \in \mathcal{A}_t$, which are rather standard assumptions). For `AdaLinUCB`, however, we do require boundedness of the noise $\varepsilon_t$ (instead of just subgaussianity). We conjecture this requirement could be dropped at the expense of a further $\log T$ factor in the regret. Finally, for efficiency reasons our implementations are not designed to capture sparsity. Hence our experiments are limited to testing the impact of model selection.

Our work leaves some open problems:

1. Proving a lower bound on the regret of sparse linear bandits when the sparsity level is unknown to the learner would be important. Citing again [17, Section 23.5], no algorithm can enjoy regret $\widetilde{O}(\sqrt{SdT})$ simultaneously for all sparsity levels $S$. However, we do not know whether the known lower bound $\Omega(\sqrt{SdT})$ can be strengthened to $\Omega(S\sqrt{dT})$ in the agnostic case.

2. Our instance-dependent regret bound is of order $\widetilde{O}\big(\max\{S^2, d\}\frac{d}{\Delta}\big)$. It would be interesting to prove an upper bound that improves on the factor $d^2/\Delta$, or a lower bound showing that $d^2/\Delta$ cannot be improved on.

3. Our bound on the regret of `AdaLinUCB` looks pessimistic due to the presence of the constant exploration probability $q$. It would be interesting to prove a bound that more closely reflects the good empirical performance of this algorithm.

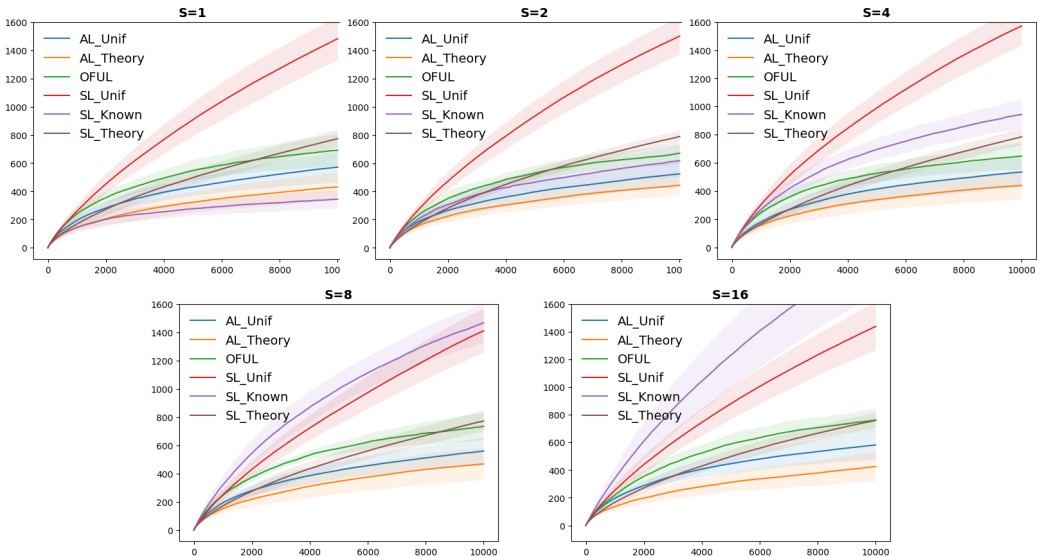

Figure 1: Experimental results with different sparsity levels $S \in \{1, 2, 4, 8, 16\}$. In each plot, the $X$-axis are time steps in $[1, 10^4]$ and the $Y$-axis is cumulative regret. `AL` stands for `AdaLinUCB` and `SL` stands for `SparseLinUCB`.

## Acknowledgments and Disclosure of Funding

We thank the anonymous reviewers for their helpful comments. This research is supported by the MUR PRIN grant 2022EKNE5K (Learning in Markets and Society), the FAIR (Future Artificial Intelligence Research) project, funded by the NextGenerationEU program within the PNRR-PE-AI scheme, the EU Horizon CL4-2022-HUMAN-02 research, innovation action under grant agreement 101120237, project ELIAS (European Lighthouse of AI for Sustainability), a Singapore Ministry of Education AcRF Tier 2 grant (A-8000423-00-00), and the National Research Foundation, Singapore under its AI Singapore Program (AISG Award No: AISG-PhD/2021-01-004[T]).

In particular, NCB and KJ acknowledge the financial support from the MUR PRIN grant, the FAIR project, and the ELIAS project. TJ is funded by a Singapore Ministry of Education AcRF Tier 2 grant (A-8000423-00-00) and the National Research Foundation, Singapore under its AI Singapore Program (AISG Award No: AISG-PhD/202101-004[T]).

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

Table 2: Notation.

| Symbol | Description |
| --- | --- |
| $\mathcal{A}_t$ | Action set at time $t$ |
| $\theta_*$ | True parameter of the linear model |
| $d$ | Dimension of $\theta_*$ |
| $S$ | Number of nonzero components in $\theta_*$ |
| $X_t$ | Random reward of pulling arm $A_t \in \mathcal{A}_t$ |
| $\widehat{X}_t$ | Prediction of SeqSEW at time $t$ |
| $\mathcal{C}_t^s$ | The confidence set with radius $2^s \log T$ and center $\widehat{\theta}_{t-1}$ |
| $I_t$ | The index of the confidence set drawn at time $t$ |
| $q_s$ | The probability $\mathbb{P}(I_t = s)$ of drawing index $s$ of the confidence set in SparseLinUCB |
| $o$ | The index of the smallest safe confidence set, defined in (3.2) |
| $Q$ | The probability $\mathbb{P}(I_t \geq o) = q_o + \cdots + q_n$ of drawing a safe confidence set in SparseLinUCB |
| $A_t^s$ | The optimistic action for $\mathcal{C}_t^s$ |
| $A_t^*$ | The optimal action in $\mathcal{A}_t$ |
| $\mathcal{E}$ | Event that $\theta_* \in \mathcal{C}_t$ for all $t \in [T]$ |
| $\det V$ | Determinant of $V$ |

## A  Notation

In Table 2 we list the most used notations. Next, we recall some definitions that are used throughout this appendix. We have $V_t = I + \sum_{s=1}^{t} A_s A_s^\top$ and

$$\widehat{\theta}_t = \underset{\theta \in \mathbb{R}^d}{\text{argmin}} \left( \|\theta\|_2^2 + \sum_{s=1}^{t} \left( \widehat{X}_s - \langle \theta, A_s \rangle \right)^2 \right)$$

where $A_s \in \mathcal{A}_s$ is the action chosen by the learner at round $t$. Recall that $\alpha_i = 2^i \log T$. For $i \in [n]$,

$$A_t^i = \underset{a \in \mathcal{A}_t}{\text{argmax}} \max_{\theta \in \mathcal{C}_t^i} \langle \theta, a \rangle = \underset{a \in \mathcal{A}_t}{\text{argmax}} \left( \langle a, \widehat{\theta}_{t-1} \rangle + \|a\|_{V_{t-1}^{-1}} \sqrt{\alpha_i} \right)$$

where

$$\mathcal{C}_t^i = \left\{ \theta \in \mathbb{R}^d : \|\theta - \widehat{\theta}_{t-1}\|_{V_{t-1}}^2 \leq \alpha_i \right\}.$$

Finally, recall definition (2.2) with $\delta = 1/T$,

$$\mathcal{C}_t = \left\{ \theta \in \mathbb{R}^d : \|\theta\|_2^2 + \sum_{s=1}^{t-1} (\widehat{X}_s - \langle A_s, \theta \rangle)^2 \leq \gamma(1/T) \right\}.$$

and recall that $\mathcal{E} = \bigcap_{t \in [T]} \{\theta_* \in \mathcal{C}_t\}$.

## B  Analysis of SparseLinUCB

**Theorem 3.2.** *The expected regret of* SparseLinUCB *run with the number of models $n$ in (3.1) and a distribution $q = \{q_s\}_{s \in [n]}$ satisfies*

$$\mathbb{E}[R_T] = O\left( (\log T) \sum_{s \geq o} \sqrt{d 2^s T q_s} + (\log T) \sqrt{S T d / Q} \right)$$

*where $Q = \sum_{s \geq o} q_s$.*

*Proof.* Lemma 3.1 implies $\mathcal{C}_t \subset \mathcal{C}_t^o$. Hence, if $\mathcal{E}$ is true and $s < o$, then

$$\langle \theta_*, A_t^s \rangle \geq \langle \widehat{\theta}_{t-1}, A_t^s \rangle - \sqrt{\alpha_o} \|A_t^s\|_{V_{t-1}^{-1}} \qquad (\theta_* \in \mathcal{C}_t \subset \mathcal{C}_t^o)$$

$$= \left( \langle \widehat{\theta}_{t-1}, A_t^s \rangle + \sqrt{\alpha_s} \|A_t^s\|_{V_{t-1}^{-1}} \right) - (\sqrt{\alpha_o} + \sqrt{\alpha_s}) \|A_t^s\|_{V_{t-1}^{-1}}$$

$$\geq \left( \langle \widehat{\theta}_{t-1}, A_t^o \rangle + \sqrt{\alpha_s} \|A_t^o\|_{V_{t-1}^{-1}} \right) - (\sqrt{\alpha_o} + \sqrt{\alpha_s}) \|A_t^s\|_{V_{t-1}^{-1}} \quad \text{(maximality of } A_t^s \text{ in } \mathcal{C}_t^s)$$

$$= \langle \widehat{\theta}_{t-1}, A_t^o \rangle + \sqrt{\alpha_o} \|A_t^o\|_{V_{t-1}^{-1}} - (\sqrt{\alpha_o} - \sqrt{\alpha_s}) \|A_t^o\|_{V_{t-1}^{-1}} - (\sqrt{\alpha_o} + \sqrt{\alpha_s}) \|A_t^s\|_{V_{t-1}^{-1}}$$

$$\geq \langle \theta_*, A_t^o \rangle - (\sqrt{\alpha_o} - \sqrt{\alpha_s}) \|A_t^o\|_{V_{t-1}^{-1}} - (\sqrt{\alpha_o} + \sqrt{\alpha_s}) \|A_t^s\|_{V_{t-1}^{-1}} \quad (\theta_* \in \mathcal{C}_t \subset \mathcal{C}_t^o)$$

$$\geq \langle \theta_*, A_t^o \rangle - (\sqrt{\alpha_o} - \sqrt{\alpha_s}) \|A_t^o\|_{V_{t-1}^{-1}} - (\sqrt{\alpha_o} + \sqrt{\alpha_s}) \|A_t^o\|_{V_{t-1}^{-1}} \quad \text{(by Lemma D.3)}$$

$$= \langle \theta_*, A_t^o \rangle - 2\sqrt{\alpha_o} \|A_t^o\|_{V_{t-1}^{-1}}$$

$$\geq \langle \theta_*, A_t^* \rangle - 3\sqrt{\alpha_o} \|A_t^o\|_{V_{t-1}^{-1}} \tag{B.1}$$

where in the first and the third inequalities, we use the fact that for any $A \in \mathbb{R}^d$,

$$\langle \theta_* - \widehat{\theta}_{t-1}, A \rangle \leq \|\theta_* - \widehat{\theta}_{t-1}\|_{V_{t-1}} \|A\|_{V_{t-1}^{-1}}, \quad \text{(Cauchy–Schwarz inequality)}$$

$$\leq \sqrt{\alpha_o} \|A\|_{V_{t-1}^{-1}} \quad \text{(due to } \theta_* \in \mathcal{C}_t^o)$$

and for the last inequality,

$$\langle \theta_*, A_t^* \rangle \leq \langle \widehat{\theta}_{t-1}, A_t^o \rangle + \sqrt{\alpha_0} \|A_t^o\|_{V_{t-1}^{-1}} \quad (\theta_* \in \mathcal{C}_t^o, \text{ maximality of } A_t^o.)$$

We can decompose the regret as follows,

$$R_T = \sum_{t=1}^T \mathbb{1}\{\mathcal{E}\} \langle \theta_*, A_t^* - A_t^{I_t} \rangle + \sum_{t=1}^T \mathbb{1}\{\mathcal{E}^c\} \langle \theta_*, A_t^* - A_t^{I_t} \rangle$$

$$= \sum_{t=1}^T \mathbb{1}\{\mathcal{E}^c\} \langle \theta_*, A_t^* - A_t^{I_t} \rangle + \sum_{t=1}^T \mathbb{1}\{I_t \geq o, \mathcal{E}\} \langle \theta_*, A_t^* - A_t^{I_t} \rangle + \sum_{t=1}^T \mathbb{1}\{I_t < o, \mathcal{E}\} \langle \theta_*, A_t^* - A_t^{I_t} \rangle$$

$$\leq \sum_{t=1}^T \mathbb{1}\{\mathcal{E}^c\} \langle \theta_*, A_t^* - A_t^{I_t} \rangle + \sum_{t=1}^T \mathbb{1}\{I_t \geq o, \mathcal{E}\} \langle \theta_*, A_t^* - A_t^{I_t} \rangle$$

$$+ \sum_{t=1}^T \mathbb{1}\{I_t < o, \mathcal{E}\} \min \left\{ 2, 3\sqrt{\alpha_o} \|A_t^o\|_{V_{t-1}^{-1}} \right\}$$

$$\leq \sum_{t=1}^T \mathbb{1}\{\mathcal{E}^c\} \langle \theta_*, A_t^* - A_t^{I_t} \rangle + \underbrace{\sum_{t=1}^T \mathbb{1}\{I_t \geq o, \mathcal{E}\} \langle \theta_*, A_t^* - A_t^{I_t} \rangle}_{\spadesuit} + \underbrace{\sum_{t=1}^T \left\{ 2, 3\sqrt{\alpha_o} \|A_t^o\|_{V_{t-1}^{-1}} \right\}}_{\clubsuit}.$$

$$\tag{B.2}$$

Since $\mathbb{P}(\mathcal{E}^c) \leq \delta \leq \frac{1}{T}$, the first sum in the above line is easily bounded,

$$\sum_{t=1}^T \mathbb{1}\{\mathcal{E}^c\} \langle \theta_*, A_t^* - A_t^{I_t} \rangle \leq 2T\mathbb{P}(\mathcal{E}^c) \leq 2 .$$

**Bounding term ♣.** For each $s \geq 0$, let

$$\mathcal{T}^s = \left\{ t \in [T] : \det(V_{t-1}) \in \left[ 2^{sd}, 2^{(s+1)d} \right) \right\}.$$

Note that $\det(V_t)$ is monotone increasing w.r.t $t$. Define $s' = \lceil \log_2 \det(V_T)/d \rceil$. Then,

$$[T] = \bigcup_{s=1}^{s'} \mathcal{T}^s.$$

Therefore,

$$\clubsuit \leq \sum_{s=1}^{s'} \sum_{t \in \mathcal{T}^s} \min \left\{ 2, 3\sqrt{\alpha_o} \|A_t^o\|_{V_{t-1}^{-1}} \right\} .$$

By applying Lemma D.5, we obtain

$$\mathbb{E}[\clubsuit] = \mathbb{E}\left[\sum_{s=1}^{s'}\sum_{t\in\mathcal{T}^s}\min\left\{2, 3\sqrt{\alpha_o}\|A_t^o\|_{V_{t-1}^{-1}}\right\}\right]$$

$$= 3\sqrt{\alpha_o}\mathbb{E}\left[\sum_{s=1}^{s'}\sum_{t\in\mathcal{T}^s}\min\left\{1, \|A_t^o\|_{V_{t-1}^{-1}}\right\}\right] \qquad\qquad (\alpha_o \geq 1)$$

$$\leq 3\sqrt{\alpha_o}\sqrt{T\cdot\mathbb{E}\left[\sum_{s=1}^{s'}\left[\sum_{t\in\mathcal{T}^s}\min\left\{1, \|A_t^o\|_{V_{t-1}^{-1}}^2\right\}\right]\right]} \qquad \text{(Cauchy–Schwarz inequality)}$$

$$\leq 3\sqrt{\alpha_o T}\sqrt{\sum_{s=1}^{s'}(2d/Q + 1/Q)} \qquad\qquad \text{(due to Lemma D.5)}$$

$$\leq 3\sqrt{\alpha_o T}\sqrt{(2d+1)s'/Q}$$

$$= 3\sqrt{\alpha_o T}\sqrt{(2d+1)/Q\lceil\log_2\det(V_T)/d\rceil}$$

$$= O\left(\log T\sqrt{STd/Q}\right).$$

**Bounding term $\spadesuit$.** Let $T_s = \sum_{t=1}^{T}\mathbb{1}\{I_t = s\}$.

$$\mathbb{E}[\spadesuit] = \mathbb{E}\left[\sum_{t=1}^{T}\mathbb{1}\{I_t \geq o, \mathcal{E}\}\langle\theta_*, A_t^* - A_t^{I_t}\rangle\right]$$

$$\leq \mathbb{E}\left[\sum_{t=1}^{T}\mathbb{1}\{I_t \geq o, \mathcal{E}\}\cdot\min\left\{2, \sqrt{\alpha_{I_t}}\|A_t^{I_t}\|_{V_{t-1}^{-1}}\right\}\right]$$

$$\leq 2\sum_{s\geq o}\mathbb{E}\left[\sqrt{\alpha_s T_s}\sqrt{\sum_{t\in[T]}\mathbb{1}\{I_t = s\}\min\left\{1, \|A_t^s\|_{V_{t-1}^{-1}}^2\right\}}\right] \qquad \text{(Cauchy–Schwarz inequality)}$$

$$\leq 2\sum_{s\geq o}\mathbb{E}\left[\sqrt{\alpha_s T_s}\sqrt{\sum_{t\in[T]}\min\left\{1, \|A_t\|_{V_{t-1}^{-1}}^2\right\}}\right]$$

$$\leq 2\sum_{s\geq o}\mathbb{E}\left[\sqrt{\alpha_s T_s}\sqrt{2\log\det V_T}\right] \qquad\qquad \text{(Lemma D.1)}$$

$$\leq 2\sum_{s\geq o}\mathbb{E}\left[\sqrt{\alpha_s T_s}\right]\cdot O(\sqrt{d\log T}) \qquad\qquad \text{(upper bound on } \det(V_T))$$

$$\leq 2\sum_{s\geq o}\sqrt{\alpha_s}\sqrt{\mathbb{E}[T_s]}\cdot O(\sqrt{d\log T}) \qquad\qquad \text{(Jensen's inequality)}$$

$$= O\left(\sum_{s\geq o}\sqrt{\alpha_s dT q_s\log T}\right) \qquad\qquad \text{(because } \mathbb{P}(I_t = s) = q_s.)$$

Substituting the bounds of $\spadesuit$ and $\clubsuit$ to (B.2), we have

$$\mathbb{E}[R_T] \leq O\left(\sum_{s\geq o}\sqrt{d\alpha_s T\log T q_s} + \log T\sqrt{SdT/Q}\right)$$

concluding the proof. $\qquad\qquad\qquad\qquad\qquad\qquad\qquad\qquad\qquad\qquad\qquad\qquad\qquad\square$

**Theorem 3.4.** *The expected regret of* `SparseLinUCB` *run with the number of models $n$ in* (3.1), *a distribution $\boldsymbol{q} = \{q_s\}_{s\in[n]}$ and using* `SeqSEW` *as base algorithm satisfies*

$$\mathbb{E}[R_T] = O\left(\frac{(dS/Q) + d^2}{\Delta}(\log T)^2\right)$$

*where $Q = \sum_{s \geq o} q_s$.*

*Proof.*

$$R_T = \sum_{t=1}^{T} \mathbb{1}\{\mathcal{E}\}\langle\theta_*, A_t^* - A_t^{I_t}\rangle + \sum_{t=1}^{T} \mathbb{1}\{\mathcal{E}^c\}\langle\theta_*, A_t^* - A_t^{I_t}\rangle$$

$$\leq \sum_{t=1}^{T} \mathbb{1}\{\mathcal{E}^c\}\langle\theta_*, A_t^* - A_t^{I_t}\rangle + \sum_{t=1}^{T} \mathbb{1}\{I_t \geq o, \mathcal{E}\}\frac{\langle\theta_*, A_t^* - A_t^{I_t}\rangle^2}{\Delta}$$

$$+ \sum_{t=1}^{T} \mathbb{1}\{I_t < o, \mathcal{E}\}\frac{\langle\theta_*, A_t^* - A_t^{I_t}\rangle^2}{\Delta} \qquad \text{(minimality of } \Delta\text{)}$$

$$\leq \sum_{t=1}^{T} \mathbb{1}\{\mathcal{E}^c\}\langle\theta_*, A_t^* - A_t^{I_t}\rangle + \underbrace{\sum_{t=1}^{T} \mathbb{1}\{I_t \geq o, \mathcal{E}\}\frac{\langle\theta_*, A_t^* - A_t^{I_t}\rangle^2}{\Delta}}_{\spadesuit}$$

$$+ 9 \underbrace{\sum_{t=1}^{T} \mathbb{1}\{I_t < o, \mathcal{E}\}\frac{\min\left\{1, \alpha_o\|A_t^o\|_{V_{t-1}^{-1}}^2\right\}}{\Delta}}_{\clubsuit} \qquad \text{(by applying (B.1) to } \clubsuit\text{)}$$

**Bounding term $\clubsuit$.** Let $s' = \lceil \log_2 \det(V_T)/d \rceil$. By applying Lemma D.5, we obtain

$$\mathbb{E}[\clubsuit] \leq \frac{9}{\Delta}\mathbb{E}\left[\sum_{s \geq 0}\sum_{t \in \mathcal{T}^s}\min\left\{1, \alpha_o\|A_t^o\|_{V_{t-1}^{-1}}^2\right\}\right]$$

$$\leq \frac{9\alpha_o}{\Delta}\sum_{s=1}^{s'}\mathbb{E}\left[\sum_{t \in \mathcal{T}^s}\min\left\{1, \|A_t^o\|_{V_{t-1}^{-1}}^2\right\}\right] \qquad (\alpha_o \geq 1)$$

$$\leq \frac{9\alpha_o}{\Delta}\sum_{s=1}^{s'}(2d/Q + 1/Q) \qquad \text{(due to Lemma D.5)}$$

$$\leq \frac{27\alpha_o ds'/Q}{\Delta}$$

$$= O\left(\frac{dS(\log T)^2/Q}{\Delta}\right) \qquad \text{(because } s' = O(\log T)\text{.)}$$

**Bounding term $\spadesuit$.**

$$\mathbb{E}[\spadesuit] = \mathbb{E}\left[\sum_{t=1}^{T}\mathbb{1}\{I_t \geq o, \mathcal{E}\}\frac{\langle\theta_*, A_t^* - A_t^{I_t}\rangle^2}{\Delta}\right]$$

$$\leq \frac{4}{\Delta}\mathbb{E}\left[\sum_{t=1}^{T}\mathbb{1}\{I_t \geq o, \mathcal{E}\}\cdot\min\left\{1, \alpha_{I_t}\|A_t^{I_t}\|_{V_{t-1}^{-1}}^2\right\}\right]$$

$$\leq 4\sum_{s \geq o}\frac{\alpha_s}{\Delta}\mathbb{E}\left[\sum_{t \in [T]}\min\left\{1, \|A_t\|_{V_{t-1}^{-1}}^2\right\}\right]$$

$$\leq 8\sum_{s \geq o}\frac{\alpha_s}{\Delta}\cdot\log\det V_T \qquad \text{(due to Lemma D.1)}$$

$$= O\left(\frac{d^2(\log T)^2}{\Delta}\right) \qquad (B.3)$$

where the first inequality comes from

$$\langle\theta_*, A_t^* - A_t^{I_t}\rangle \leq \max_{\theta \in \mathcal{C}_{t-1}^{I_t}}\langle\theta - \theta_*, A_t^{I_t}\rangle$$

$$\leq \max_{\theta \in \mathcal{C}_{t-1}^{I_t}} \|\theta - \theta_*\|_{V_{t-1}} \|A_t^{I_t}\|_{V_{t-1}^{-1}} \qquad \text{(Cauchy-Schwarz Inequality)}$$

$$\leq 2\sqrt{\alpha_{I_t}} \|A_t^{I_t}\|_{V_{t-1}^{-1}}$$

where the last inequality holds because $\mathcal{E}$ and $I_t \geq o$ both hold, and so $\theta_* \in \mathcal{C}_{t-1} \subset \mathcal{C}_{t-1}^{I_t}$ due to Lemma 3.1. The factor 2 is due to an application of the triangular inequality. Substituting the bounds of ♠ and ♣ in (B.2), we have

$$\mathbb{E}[R_T] = O\left( \max\{S/Q, d\} \cdot \frac{d(\log T)^2}{\Delta} \right).$$

$\square$

**Corollary B.1.** *Pick any $C \geq 1$ and let $\{q_s\}_{s \in [n]}$ be chosen as in (3.4). Then the expected regret of* `SparseLinUCB` *is*

$$\mathbb{E}[R_T] = \widetilde{O}\left( \min\left\{ \max\left\{C, S/C\right\}\sqrt{dT}, \frac{\max\left\{d^2, S^2 d/C^2\right\}}{\Delta} \right\} \right)$$

*Proof.* We consider the following cases based on the value of $C$.
**Case 1:** $C^2 < 2^o$.
$q_o = C^2 \cdot 2^{-o}$. According to Theorem 3.2,

$$\mathbb{E}[R_T] = O\left( (\log T) \sum_{s \geq o} \sqrt{d2^s T q_s} + (\log T)\sqrt{STd/Q} \right)$$

$$\leq O\left( (\log T)n\sqrt{d2^o T q_o} + (\log T)\sqrt{STd/q_o} \right)$$

$$= \widetilde{O}\left( \max\{C, S/C\}\sqrt{dT} \right). \qquad \text{(due to } n = O(\log d) \text{ and } 2^o = \Theta(S \log T))$$

Besides, according to Theorem 3.4,

$$\mathbb{E}[R_T] = O\left( \max\{S/Q, d\} \cdot \frac{d \log^2 T}{\Delta} \right)$$

$$= O\left( \max\{S/q_o, d\} \cdot \frac{d \log^2 T}{\Delta} \right)$$

$$= \widetilde{O}\left( \frac{\max\{d^2, S^2 d/C^2\}}{\Delta} \right).$$

Therefore,

$$\mathbb{E}[R_T] = \widetilde{O}\left( \min\left\{ \max\left\{C, S/C\right\}\sqrt{dT}, \frac{\max\left\{d^2, S^2 d/C^2\right\}}{\Delta} \right\} \right).$$

**Case 2:** $C^2 \in [2^o, 2^n)$.
For $s$ with $C^2 < 2^s$, $q_s = C^2 2^{-s}$. Let $o' = \arg\min_{s > o}\{C^2 < 2^s\}$. Then, $q_{o'} \geq 1/4$. According to Theorem 3.2,

$$\mathbb{E}[R_T] = O\left( (\log T) \sum_{s \geq o} \sqrt{d2^s T q_s} + (\log T)\sqrt{STd/Q} \right)$$

$$\leq O\left( (\log T) \sum_{s \in [o, o')} \sqrt{d2^s T} + (\log T) \sum_{s \in [o', n]} \sqrt{d2^s T q_s} + (\log T)\sqrt{STd/q_{o'}} \right)$$

$$\leq \widetilde{O}\left( \sqrt{d2^{o'} T} + nC\sqrt{dT} + \sqrt{STd} \right)$$

$$= \tilde{O}\Big( \max\{C, S/C\}\sqrt{dT} \Big). \qquad\qquad \text{(due to } C^2 = \Theta(2^{o'}) \text{ and } 2^{o'} \geq 2^o \geq S)$$

Besides, according to Theorem 3.4,

$$\mathbb{E}[R_T] = O\Big( \max\{S/Q, d\} \cdot \frac{d(\log T)^2}{\Delta} \Big)$$

$$= O\Big( \max\{S/q_{o'}, d\} \cdot \frac{d(\log T)^2}{\Delta} \Big)$$

$$= \tilde{O}\Big( \frac{\max\{d^2, S^2 d/C^2\}}{\Delta} \Big).$$

Therefore,

$$\mathbb{E}[R_T] = \tilde{O}\left( \min\left\{ \max\left\{C, S/C\right\}\sqrt{dT}, \frac{\max\left\{d^2, S^2 d/C^2\right\}}{\Delta} \right\} \right).$$

**Case $C^2 \geq 2^n$:** $q_s = 1/n$ for all $s \in [n]$. It is easy to verify that $\mathbb{E}[R_T] = \tilde{O}(d\sqrt{T})$ and $\mathbb{E}[R_T] = \tilde{O}(d^2/\Delta)$. Thus,

$$\mathbb{E}[R_T] = \tilde{O}\left( \min\left\{ \max\left\{C, S/C\right\}\sqrt{dT}, \frac{\max\left\{d^2, S^2 d/C^2\right\}}{\Delta} \right\} \right).$$

$\square$

**Corollary B.2.** *Assume that the sparsity level $S$ is known and choose a distribution $\boldsymbol{q} = \{q_s\}_{s\in[n]}$ with $q_o = 1$, where $o$ is set as in (3.2). Then, the expected regret of* `SparseLinUCB` *is $\mathbb{E}[R_T] = \tilde{O}\big(\frac{Sd}{\Delta}\big)$.*

*Proof.* We follow the same steps as in the proof of Theorem 3.4. The only difference lies in the bounding term $\spadesuit$ in (B.3). We have

$$\mathbb{E}[\spadesuit] = \mathbb{E}\left[ \sum_{t=1}^{T} \mathbb{1}\{I_t \geq o, \mathcal{E}\} \frac{\langle \theta_*, A_t^* - A_t^{I_t}\rangle^2}{\Delta} \right]$$

$$= \mathbb{E}\left[ \sum_{t=1}^{T} \mathbb{1}\{\mathcal{E}\} \frac{\langle \theta_*, A_t^* - A_t\rangle^2}{\Delta} \right]$$

$$\leq \frac{4\alpha_o}{\Delta} \mathbb{E}\left[ \sum_{t\in[T]} \min\left\{1, \|A_t\|_{V_{t-1}^{-1}}^2\right\} \right]$$

$$= O\left( \frac{Sd(\log T)^2}{\Delta} \right) \qquad\qquad\qquad\qquad\qquad \text{(B.4)}$$

where the second equality is because $q_o = 1$ and so $A_t^{I_t} = A_t^o = A_t$, and the first inequality comes from

$$\langle \theta_*, A_t^* - A_t\rangle \leq \max_{\theta\in\mathcal{C}_{t-1}^o} \langle \theta - \theta_*, A_t\rangle$$

$$\leq \max_{\theta\in\mathcal{C}_{t-1}^o} \|\theta - \theta_*\|_{V_{t-1}} \|A_t^o\|_{V_{t-1}^{-1}} \qquad\qquad \text{(Cauchy-Schwarz Inequality)}$$

$$\leq 2\sqrt{\alpha_o}\|A_t\|_{V_{t-1}^{-1}}$$

where the last inequality holds because $\mathcal{E}$ and $I_t = o$ both hold, and so $\theta_* \in \mathcal{C}_{t-1} \subset \mathcal{C}_{t-1}^o$ due to Lemma 3.1. Substituting the bounds of $\spadesuit$ and $\clubsuit$ in (B.2), we have

$$\mathbb{E}[R_T] = O\left( \frac{Sd(\log T)^2}{\Delta} \right).$$

$\square$

## C  Analysis of `AdaLinUCB`

**Theorem 4.1.** *If the random independent noise $\varepsilon_t$ in (2.1) satisfies $\varepsilon_t \in [-1, 1]$ for all $t \in [T]$, then the regret of* `AdaLinUCB` *run with $\eta = \sqrt{(\log n)/(Tn)}$ for $n$ in (3.1) and $q \in (0, 1]$ satisfies*

$$\mathbb{E}[R_T] \leq \left( \sqrt{8\alpha_n q} + 4\sqrt{(2\alpha_o)/q} \right) \sqrt{dT \log\left( 1 + \frac{TL^2}{d} \right) + 1} + O\left( \sqrt{nT \log n} \right)$$

$$= \tilde{O}\left( \max\left\{ \sqrt{dq}, \sqrt{S/q} \right\} \sqrt{dT} \right) .$$

*Proof.* Let $\mathcal{T}_1$ be the set of time steps where $Z_t = 1$ in Line 4 of `AdaLinUCB` and let $\mathcal{T}_2 = [T] \setminus \mathcal{T}_1$. For all $t \in \mathcal{T}_2$ the action $A_t \in \mathcal{A}_t$ is chosen by `Exp3` through an adversarial mapping $\mu_t : [n] \to \mathcal{A}_t$ defined in Lines 9–10. The resulting reward $X_t \in [-2, 2]$ is fed to `Exp3` as a $[0, 1]$-valued loss $\ell_t(I_t) = (2 - X_t)/4$, where $\mu_t(I_t) = A_t$. Since $\mathcal{T}_2$ is selected through independent coin tosses with fixed bias $q$, we can apply the `Exp3` regret analysis (for losses chosen by a non-oblivious adversary) to bound the regret in $\mathcal{T}_2$ and show that

$$\sum_{t \in \mathcal{T}_2} \langle A_t^o - A_t, \theta_* \rangle = 4 \sum_{t \in \mathcal{T}_2} \left( \ell_t(I_t) - \ell_t(o) \right) = O\left( \sqrt{n \log n} \right) \tag{C.1}$$

where we used $\mu_t(o) = A_t^o$ for all $t \in [T]$.

We have

$$R_{\mathcal{T}_2} = \sum_{t \in \mathcal{T}_2} \operatorname*{argmax}_{a \in \mathcal{A}_t} \langle a - A_t, \theta_* \rangle$$

$$= \sum_{t \in \mathcal{T}_2} \operatorname*{argmax}_{a \in \mathcal{A}_t} \langle a - A_t^o + A_t^o - A_t, \theta_* \rangle$$

$$= \underbrace{\sum_{t \in \mathcal{T}_2} \operatorname*{argmax}_{a \in \mathcal{A}_t} \langle a - A_t^o, \theta_* \rangle}_{\text{Reg}_{\text{Img}}} + \underbrace{\sum_{t \in \mathcal{T}_2} \langle A_t^o - A_t, \theta_* \rangle}_{\text{Reg}_{\text{Exp3}}} .$$

$\text{Reg}_{\text{Exp3}}$ is bounded in (C.1), hence we only need to bound

$$\text{Reg}_{\text{Img}} = \sum_{t \in \mathcal{T}_2} \operatorname*{argmax}_{a \in \mathcal{A}_t} \langle a - A_t^o, \theta_* \rangle . \tag{C.2}$$

Let

$$\widetilde{\theta}_t^i = \operatorname*{argmax}_{\theta \in \mathcal{C}_t^i} \max_{a \in \mathcal{A}_t} \langle a, \theta \rangle \quad \text{and} \quad A_t^* = \operatorname*{argmax}_{a \in \mathcal{A}_t} \langle a, \theta_* \rangle .$$

Recall $\mathcal{E}$ is the event that $\theta_* \in \mathcal{C}_t$ for all $t \in [T]$. Assume $\mathcal{E}$ holds. We obtain that for $t \in [T]$,

$$\langle \theta_*, A_t^* - A_t^o \rangle \leq \langle \widetilde{\theta}_t^o - \theta_*, A_t^o \rangle \qquad \text{(because } \theta_* \in \mathcal{C}_t \subseteq \mathcal{C}_t^o\text{)}$$

$$\leq \|\widetilde{\theta}_t^o - \theta_*\|_{V_{t-1}} \|A_t^o\|_{V_{t-1}^{-1}} \qquad \text{(Cauchy–Schwarz inequality)}$$

$$\leq \sqrt{\alpha_o} \|A_t^o\|_{V_{t-1}^{-1}}, \tag{C.3}$$

where the last inequality is because $\widetilde{\theta}_t^o \in \mathcal{C}_t^o$. Then, assuming $\mathcal{E}$ holds, we can bound $\text{Reg}_{\text{Img}}$ as

$$\text{Reg}_{\text{Img}} = \sum_{t \in \mathcal{T}_2} \langle \theta_*, A_t^* - A_t^o \rangle$$

$$\leq \sum_{t \in \mathcal{T}_2} \min\left\{ 2, \sqrt{\alpha_o} \|A_t^o\|_{V_{t-1}^{-1}} \right\}$$

$$\leq \sum_{t \in \mathcal{T}_2} \min\left\{ 2, \sqrt{2\gamma(1/T)} \|A_t^o\|_{V_{t-1}^{-1}} \right\} \qquad \text{(due to } \alpha_o \leq 2\gamma(1/T)\text{)}$$

$$\leq 2\sqrt{\gamma(1/T)} \sum_{t \in \mathcal{T}_2} \min\left\{ 1, \|A_t^o\|_{V_{t-1}^{-1}} \right\}, \tag{C.4}$$

where in the first inequality, we use the facts $\langle \theta_*, A_t^* - A_t^o \rangle \leq \sqrt{\alpha_o} \|A_t^o\|_{V_{t-1}^{-1}}$ and $\langle \theta_*, A_t^* - A_t^o \rangle \leq \langle \theta_*, A_t^* \rangle \leq 2$.

From Lemma D.3, we have

$$\sum_{t \in \mathcal{T}_2} \min \left\{ 1, \|A_t^o\|_{V_{t-1}^{-1}}^2 \right\} \leq \sum_{t \in \mathcal{T}_2} \min \left\{ 1, \|A_t^n\|_{V_{t-1}^{-1}}^2 \right\}.$$

For each $s \geq 0$, let

$$\mathcal{T}_2^s = \left\{ t \in \mathcal{T}_2 : \det(V_{t-1}) \in \left[ 2^{ds}, 2^{d(s+1)} \right) \right\}$$

so that

$$\mathcal{T}_2 = \bigcup_{s \geq 0} \mathcal{T}_2^s.$$

Let $s' = \lceil \log_2 \det(V_T)/d \rceil$. We have

$$\mathbb{E}\left[ \mathrm{Reg}_{\mathrm{Img}} \cdot \mathbb{1}[\mathcal{E}] \right] \leq \mathbb{E}\left[ 2\sqrt{\gamma(1/T)} \sum_{t \in \mathcal{T}_2} \min \left\{ 1, \|A_t^o\|_{V_{t-1}^{-1}} \right\} \right] \qquad \text{(due to (C.4))}$$

$$\leq 2\sqrt{2\alpha_o}\, \mathbb{E}\left[ \sum_{t \in \mathcal{T}_2} \min \left\{ 1, \|A_t^o\|_{V_{t-1}^{-1}} \right\} \right] \qquad \text{(due to } \alpha_0 \leq 2\gamma(1/T))$$

$$\leq 2\sqrt{2\alpha_o T}\sqrt{\mathbb{E}\left[ \sum_{t \in \mathcal{T}_2} \min \left\{ 1, \|A_t^o\|_{V_{t-1}^{-1}}^2 \right\} \right]} \qquad \text{(Cauchy–Schwarz inequality)}$$

$$\leq 2\sqrt{2\alpha_o T}\sqrt{\sum_{s=1}^{s'} \mathbb{E}\left[ \sum_{t \in \mathcal{T}_2^s} \min \left\{ 1, \|A_t^o\|_{V_{t-1}^{-1}}^2 \right\} \right]}$$

$$\text{(where } d \ln s' \leq \log \det(V_T))$$

$$\leq 2\sqrt{2\alpha_o T}\sqrt{\sum_{s=1}^{s'} \mathbb{E}\left[ \sum_{t \in \mathcal{T}_2^s} \min \left\{ 1, \|A_t^n\|_{V_{t-1}^{-1}}^2 \right\} \right]}$$

$$\text{(because } \|A_t^n\|_{V_{t-1}^{-1}} \geq \|A_t^o\|_{V_{t-1}^{-1}})$$

$$\leq 2\sqrt{2\alpha_o T}\sqrt{\sum_{s=1}^{s'} (2d+1)/q} \qquad \text{(due to Lemma D.4)}$$

$$\leq 2\sqrt{2(2d+1)\alpha_o T s'/q}.$$

To obtain the final results, we have

$$\det(V_T) = \prod_{i=1}^{d} \lambda_i \leq \left( \frac{1}{d}\mathrm{trace}(V_T) \right)^d \leq \left( 1 + \frac{TL^2}{d} \right)^d, \qquad \text{(C.5)}$$

where $\lambda_1, \cdots, \lambda_d$ are the eigenvalues of $V_T$. Therefore, we have $s' \leq \lceil \log_2(1 + TL^2/d) \rceil$.

$$\mathbb{E}\left[ \mathrm{Reg}_{\mathrm{Img}} \cdot \mathbb{1}[\mathcal{E}] \right] \leq 2\sqrt{2(2d+1)\alpha_o T/q} \cdot \sqrt{s'}$$

$$\leq 2\sqrt{(6d\alpha_o T/q)\lceil \log_2(1 + TL^2/d) \rceil}.$$

By substituting the bounds on $\mathrm{Reg}_{\mathrm{Img}}$ and $\mathrm{Reg}_{\mathrm{Exp3}}$ into $R_{\mathcal{T}_2}$, we obtain

$$\mathbb{E}[R_{\mathcal{T}_2}] = \mathbb{E}[\mathrm{Reg}_{\mathrm{Exp3}}\mathbb{1}\{\mathcal{E}\}] + \mathbb{E}[\mathrm{Reg}_{\mathrm{Img}}] + T \cdot \mathbb{P}(\mathcal{E}^c)$$

$$\leq 2\sqrt{6d\alpha_o T/q}\sqrt{\lceil \log(1 + TL^2/d) \rceil} + O(\sqrt{nT \log n}),$$

where the last inequality is because $\mathbb{P}(\mathcal{E}^c) \leq 1/T$ from Lemma 2.1 with our choice of $\delta = 1/\delta$.

Finally, we bound $R_{\mathcal{T}_1}$. Conditioned on event $\mathcal{E}$ and using (D.3), $\forall t \in [T], \theta_* \in \mathcal{C}_t \subseteq \mathcal{C}_t^0 \subseteq \mathcal{C}_t^n$. Hence, we can obtain

$$
\begin{aligned}
\langle \theta_*, A_t^* - A_t^n \rangle &\le \langle \widetilde{\theta}_t^n - \theta_*, A_t^n \rangle & \text{(because } \theta_* \in \mathcal{C}_t \subseteq \mathcal{C}_t^n\text{)} \\
&\le \|\widetilde{\theta}_t^n - \theta_*\|_{V_{t-1}} \|A_t^n\|_{V_{t-1}^{-1}} & \text{(Cauchy–Schwarz inequality)} \\
&\le \sqrt{\alpha_n} \|A_t^n\|_{V_{t-1}^{-1}} & \text{(because } \widetilde{\theta}_t^n \in \mathcal{C}_t^n.\text{)}
\end{aligned}
$$

Hence, conditioned on event $\mathcal{E}$,

$$
\begin{aligned}
R_{\mathcal{T}_1} &\le \sum_{t \in \mathcal{T}_1} \min \left\{ 2, \langle \theta_*, A_t^* - A_t^n \rangle \right\} \\
&\le \sum_{t \in \mathcal{T}_1} \min \left\{ 2, \sqrt{\alpha_n} \|A_t^n\|_{V_{t-1}^{-1}} \right\} & \text{(C.6)} \\
&\le 2\sqrt{\alpha_n |\mathcal{T}_1|} \sqrt{\sum_{t \in \mathcal{T}_1} \min \left\{ 1, \|A_t^n\|_{V_{t-1}^{-1}}^2 \right\}} & \text{(Cauchy–Schwarz inequality)} \\
&\le 2\sqrt{\alpha_n |\mathcal{T}_1|} \sqrt{\sum_{t \in \mathcal{T}_1} \min \left\{ 1, \|A_t^n\|_{V_{t-1}^{-1}}^2 \right\} + \sum_{t \in \mathcal{T}_2} \min \left\{ 1, \|A_t\|_{V_{t-1}^{-1}}^2 \right\}} \\
&= 2\sqrt{\alpha_n |\mathcal{T}_1|} \sqrt{\sum_{t \in [T]} \min \left\{ 1, \|A_t\|_{V_{t-1}^{-1}}^2 \right\}} \\
&\le 2\sqrt{\alpha_n |\mathcal{T}_1|} \cdot \sqrt{2 \log(\det(V_T))}, & \text{(C.7)}
\end{aligned}
$$

where the last inequality is due to Lemma D.1. Therefore, the expected regret for $t \in \mathcal{T}_1$ is

$$
\begin{aligned}
\mathbb{E}[R_{\mathcal{T}_1}] &\le \mathbb{E}[R_{\mathcal{T}_1} \mathbb{1}\{\mathcal{E}\}] + T \cdot \mathbb{P}(\mathcal{E}^c) \\
&\le \mathbb{E}\left[ 2\sqrt{\alpha_n |\mathcal{T}_1|} \cdot \sqrt{2 \log(\det(V_T))} \right] + 1 \\
&\le \mathbb{E}\left[ 2\sqrt{\alpha_n |\mathcal{T}_1|} \cdot \sqrt{2d \log \left( 1 + \frac{TL^2}{d} \right)} \right] + 1 \\
&= 2\sqrt{\alpha_n} \mathbb{E}\left[ \sqrt{|\mathcal{T}_1|} \right] \cdot \sqrt{2d \log \left( 1 + \frac{TL^2}{d} \right)} + 1 \\
&\le 2\sqrt{\alpha_n T q} \cdot \sqrt{2d \log \left( 1 + \frac{TL^2}{d} \right)} + 1, & \text{(Jensen's inequality)}
\end{aligned}
$$

where the first inequality is due to (C.7) and $\mathbb{P}(\mathcal{E}^c) \le 1/T$ from Lemma 2.1 and the last inequality is due to the fact that for any $t \in [T]$, with probability $q$, $t \in \mathcal{T}_1$. Finally, we obtain

$$
\begin{aligned}
\mathbb{E}[R_T] &= \mathbb{E}[R_{\mathcal{T}_1}] + \mathbb{E}[R_{\mathcal{T}_2}] \\
&\le 2\sqrt{\alpha_n T q} \cdot \sqrt{2d \log \left( 1 + \frac{TL^2}{d} \right)} + 2\sqrt{6 d \alpha_o T/q} \sqrt{\lceil \log(1 + TL^2/d) \rceil} + O(\sqrt{nT \log n}).
\end{aligned}
$$

Note that $n = \Theta(\log d)$ and $\alpha_o = \Theta(S \log T)$. We obtain

$$
\mathbb{E}[R_T] = O\left( \max \left\{ \sqrt{q d}, \sqrt{\frac{S}{q}} \right\} \cdot \sqrt{dT} \cdot \log T \right),
$$

which completes the proof. $\qquad\square$

# D Supporting lemmas

**Lemma D.1** (Dani et al. [10]). *Let $A_1, A_2, \ldots, A_T \in \mathbb{R}^d$ and $V_t = I + \sum_{s=1}^{t} A_s A_s^\top$ for all $t \in [T]$. Then*

$$\sum_{t=1}^{T} \min\left\{1, \|A_t\|_{V_{t-1}^{-1}}^2\right\} \leq 2 \log \det(V_T). \tag{D.1}$$

*Moreover,*

$$\min\left\{1, \|A_t\|_{V_{t-1}^{-1}}^2\right\} \leq 2 \log\left(1 + \|A_t\|_{V_{t-1}^{-1}}^2\right) = 2 \log\left(\frac{\det(V_t)}{\det(V_{t-1})}\right).$$

**Lemma D.2.** *For $\mathcal{C}_t$ defined in (2.2), we have that $\mathcal{C}_t \subseteq \mathcal{C}_t^o$ for all $t \in [T]$.*

*Proof.* Recall

$$\mathcal{C}_{t+1} = \left\{\theta : \|\theta\|_2^2 + \sum_{s=1}^{t}\left(\widehat{X}_s - \langle \theta, A_s \rangle\right)^2 \leq \gamma(1/T)\right\}.$$

Note that

$$\|\theta\|_2^2 + \sum_{s=1}^{t}\left(\widehat{X}_s - \langle \theta, A_s \rangle\right)^2 - \|\widehat{\theta}_t\|_2^2 - \sum_{s=1}^{t}\left(\widehat{X}_s - \langle \widehat{\theta}_t, A_s \rangle\right)^2$$

$$= \|\theta\|_2^2 - 2\theta^\top\left(\sum_{s=1}^{t}\widehat{X}_s A_s\right) + \theta^\top\left(\sum_{s=1}^{t}A_s A_s^\top\right)\theta$$

$$\quad - \|\widehat{\theta}_t\|_2^2 + 2\widehat{\theta}_t^\top\left(\sum_{s=1}^{t}\widehat{X}_s A_s\right) - \widehat{\theta}_t^\top\left(\sum_{s=1}^{t}A_s A_s^\top\right)\widehat{\theta}_t$$

$$= \|\theta\|_{V_t}^2 + 2(\widehat{\theta}_t - \theta)^\top V_t \widehat{\theta}_t - \|\widehat{\theta}_t\|_{V_t}^2 \qquad (V_t = I + \textstyle\sum_{s=1}^{t} A_s A_s^\top, \widehat{\theta}_t = V_t^{-1}\textstyle\sum_{s=1}^{t} A_s \widehat{X}_s)$$

$$= \|\theta\|_{V_t}^2 - 2\theta^\top V_t \widehat{\theta}_t + \|\widehat{\theta}_t\|_{V_t}^2 = \|\theta - \widehat{\theta}_t\|_{V_t}^2.$$

Hence, we can express the ellipsoid as

$$\mathcal{C}_{t+1} = \left\{\|\theta - \widehat{\theta}_t\|_{V_t}^2 + \|\widehat{\theta}_t\|_2^2 + \sum_{s=1}^{t}\left(\widehat{X}_s - \langle \widehat{\theta}_t, A_s \rangle\right)^2 \leq \gamma(1/T)\right\}. \tag{D.2}$$

Therefore, for all $t \geq 0$,

$$\mathcal{C}_{t+1} = \left\{\theta : \|\theta - \widehat{\theta}_t\|_{V_t}^2 + \|\widehat{\theta}_t\|_2^2 + \sum_{s=1}^{t}\left(\widehat{X}_s - \langle \widehat{\theta}_t, A_s \rangle\right)^2 \leq \gamma(1/T)\right\} \qquad \text{(due to (D.2))}$$

$$\subseteq \left\{\theta : \|\theta - \widehat{\theta}_t\|_{V_t}^2 \leq \gamma(1/T)\right\}$$

$$\subseteq \left\{\theta : \|\theta - \widehat{\theta}_t\|_{V_t}^2 \leq \alpha_o\right\} \qquad \text{(by definition of } \alpha_0\text{)}$$

$$= \mathcal{C}_{t+1}^o \tag{D.3}$$

concluding the proof. $\qquad\square$

**Lemma D.3.** *For any $1 \leq p \leq q \leq n$ and $t \in [T]$,*

$$\|A_t^p\|_{V_{t-1}^{-1}} \leq \|A_t^q\|_{V_{t-1}^{-1}}.$$

*Proof.* Note that

$$A_t^p = \operatorname*{argmax}_{a \in \mathcal{A}_t} \max_{\theta \in \mathcal{C}_t^p} \langle \theta, a \rangle = \operatorname*{argmax}_{a \in \mathcal{A}_t} \langle a, \widehat{\theta}_{t-1} \rangle + \sqrt{\alpha_p}\|a\|_{V_{t-1}^{-1}},$$

$$A_t^q = \operatorname*{argmax}_{a \in \mathcal{A}_t} \max_{\theta \in \mathcal{C}_t^q} \langle \theta, a \rangle = \operatorname*{argmax}_{a \in \mathcal{A}_t} \langle a, \widehat{\theta}_{t-1} \rangle + \sqrt{\alpha_q}\|a\|_{V_{t-1}^{-1}}.$$

For contradiction, we assume $\|A_t^q\|_{V_{t-1}^{-1}} < \|A_t^p\|_{V_{t-1}^{-1}}$. Since $\|A_t^q\|_{V_{t-1}^{-1}} < \|A_t^p\|_{V_{t-1}^{-1}}$, $\|A_t^q\|_{V_{t-1}^{-1}} \neq \|A_t^p\|_{V_{t-1}^{-1}}$. Besides, according to the definition of $A_t^p$ and $A_t^q$, we have

$$\langle A_t^q, \widehat{\theta}_{t-1}\rangle + \sqrt{\alpha_p}\|A_t^q\|_{V_{t-1}^{-1}} \leq \langle A_t^p, \widehat{\theta}_{t-1}\rangle + \sqrt{\alpha_p}\|A_t^p\|_{V_{t-1}^{-1}} \tag{D.4}$$

$$< \langle A_t^p, \widehat{\theta}_{t-1}\rangle + \sqrt{\alpha_q}\|A_t^p\|_{V_{t-1}^{-1}} \tag{D.5}$$

$$\leq \langle A_t^q, \widehat{\theta}_{t-1}\rangle + \sqrt{\alpha_q}\|A_t^q\|_{V_{t-1}^{-1}}, \tag{D.6}$$

where the first inequality is due to the definition of $A_t^p$, the second inequality is due to $\sqrt{\alpha_p} < \sqrt{\alpha_q}$, and the last inequality is due to the definition of $A_t^q$. From the above results, we further have

$$\langle A_t^q - A_t^p, \widehat{\theta}_{t-1}\rangle \leq \sqrt{\alpha_p}\big(\|A_t^p\|_{V_{t-1}^{-1}} - \|A_t^q\|_{V_{t-1}^{-1}}\big) \qquad \text{(Due to (D.4))}$$

$$< \sqrt{\alpha_q}\big(\|A_t^p\|_{V_{t-1}^{-1}} - \|A_t^q\|_{V_{t-1}^{-1}}\big)$$
$$\text{(Due to assumption } \|A_t^q\|_{V_{t-1}^{-1}} < \|A_t^p\|_{V_{t-1}^{-1}} \text{ and } \alpha_q > \alpha_p\text{)}$$

$$\leq \langle A_t^q - A_t^p, \widehat{\theta}_{t-1}\rangle. \qquad \text{(Due to (D.6).)}$$

We obtain a contradiction. Therefore,

$$\|A_t^q\|_{V_{t-1}^{-1}} \geq \|A_t^p\|_{V_{t-1}^{-1}}. \tag{D.7}$$

$\square$

**Lemma D.4.**

$$\mathbb{E}\left[\sum_{t\in\mathcal{T}_2^s} \min\left\{1, \|A_t^n\|_{V_{t-1}^{-1}}^2\right\}\right] \leq 2d/q + 1/q.$$

*Proof.* Let $\det(V_{t-1}) = (q_{t-1})^d$ and $\det(V_{t-1} + A_t^n(A_t^n)^\top) = (q_{t-1} + x_t)^d$. For each $s \geq 0$, let

$$\mathcal{T}^s = \left\{t \in [T] : \det(V_{t-1}) \in \left[2^{sd}, 2^{d(s+1)}\right)\right\}.$$

Since $\mathcal{T}_2^s \subseteq \mathcal{T}^s$, to show

$$\mathbb{E}\left[\sum_{t\in\mathcal{T}_2^s} \min\left\{1, \|A_t^n\|_{V_{t-1}^{-1}}^2\right\}\right] \leq 2d/q + 1/q,$$

we only need to prove

$$\mathbb{E}\left[\sum_{t\in\mathcal{T}^s} \min\left\{1, \|A_t^n\|_{V_{t-1}^{-1}}^2\right\}\right] \leq 2d/q + 1/q.$$

We let $I_t = 1$ if the coin tosses in the $t$'s round is Head and 0 otherwise. We divide $\mathcal{T}^s$ into two disjoint parts $\underline{\mathcal{T}^s}$ and $\overline{\mathcal{T}^s}$. Specifically,

- for $\underline{\mathcal{T}^s}$, it holds that for $t \in \underline{\mathcal{T}^s}$, $\det(V_{t-1} + A_t^n(A_t^n)^\top) \leq 2^{d(s+1)}$.

- for $\overline{\mathcal{T}^s}$, it holds that for $t \in \overline{\mathcal{T}^s}$, $\det(V_{t-1} + A_t^n(A_t^n)^\top) > 2^{d(s+1)}$.

From definition of $\underline{\mathcal{T}^s}$ and the fact that if $I_t = 1$, $V_t = V_{t-1} + A_t^n(A_t^n)^\top$, we have

$$\sum_{t\in\underline{\mathcal{T}^s}} x_t \cdot I_t \leq 2^s.$$

From Algorithm 2, with probability $q$, $I_t = 1$. Therefore, if we let $\{\mathcal{F}_t\}_{t\in[T]}$ be the natural filtration of $\{A_t, I_t, X_t\}_{t\in[T]}$, we have

$$2^s \geq \mathbb{E}\left[\sum_{t\in\underline{\mathcal{T}^s}} x_t \cdot I_t\right]$$

$$= \mathbb{E}\left[\sum_{t\in[T]} x_t \cdot I_t \cdot \mathbb{1}\{t \in \underline{\mathcal{T}^s}\}\right]$$

$$= \sum_{t\in[T]} \mathbb{E}\left[x_t \cdot I_t \cdot \mathbb{1}\{t \in \underline{\mathcal{T}^s}\}\right]$$

$$= \sum_{t\in[T]} \mathbb{E}\left[\mathbb{E}\left[x_t \cdot I_t \cdot \mathbb{1}\{t \in \underline{\mathcal{T}^s}\}|\mathcal{F}_{t-1}\right]\right] \qquad \text{(Law of total expectation)}$$

$$= \sum_{t\in[T]} \mathbb{E}\left[x_t \cdot \mathbb{1}\{t \in \underline{\mathcal{T}^s}\} \cdot \mathbb{E}\left[I_t|\mathcal{F}_{t-1}\right]\right] \qquad \text{($x_t$ and $\mathbb{1}\{t \in \underline{\mathcal{T}^s}\}$ are $\mathcal{F}_{t-1}$-measurable)}$$

$$= \sum_{t\in[T]} \mathbb{E}\left[x_t \cdot \mathbb{1}\{t \in \underline{\mathcal{T}^s}\} \cdot q\right] \qquad \text{($I_t$ is an independent coin-tossing)}$$

$$= q\mathbb{E}\left[\sum_{t\in\underline{\mathcal{T}^s}} x_t\right]$$

and therefore $\mathbb{E}\left[\sum_{t\in\underline{\mathcal{T}^s}} x_t\right] \le 2^s/q$. For $t \in \underline{\mathcal{T}^s}$, we further have

$$\min\left\{1, \|A_t^n\|_{V_{t-1}^{-1}}^2\right\} \le 2\log\left(\frac{\det(V_{t-1} + A_t^n(A_t^n)^\top)}{\det(V_{t-1})}\right) \qquad \text{(due to Lemma D.1)}$$

$$= 2d\log\left(1 + \frac{x_t}{q_{t-1}}\right)$$

$$\le 2d\log\left(1 + \frac{x_t}{2^s}\right) \qquad \text{(since $t \in \mathcal{T}^s$, $q_{t-1} \ge 2^s$)}$$

$$\le \frac{2d \cdot x_t}{2^s} \qquad \text{(due to $\log(1+x) \le x$ for $x > 0$.)}$$

Therefore,

$$\mathbb{E}\left[\sum_{t\in\underline{\mathcal{T}^s}} \min\left\{1, \|A_t^n\|_{V_{t-1}^{-1}}^2\right\}\right] \le \mathbb{E}\left[\sum_{t\in\underline{\mathcal{T}^s}} \frac{2d \cdot x_t}{2^s}\right] = \frac{2d}{2^s}\mathbb{E}\left[\sum_{t\in\underline{\mathcal{T}^s}} x_t\right] \le 2d/q.$$

From definition of $\overline{\mathcal{T}^s}$, if $I_t = 1$ and $t \in \overline{\mathcal{T}^s}$, $\det(V_t) > 2^{d(s+1)}$. Then, for all $\tau > t$, $\tau \notin \mathcal{T}^s$. Therefore, there is at most one $t \in \overline{\mathcal{T}^s}$ with $I_t = 1$. We obtain

$$\mathbb{E}\left[\sum_{t\in\overline{\mathcal{T}^s}} \min\left\{1, \|A_t^n\|_{V_{t-1}^{-1}}^2\right\}\right] \le \mathbb{E}\left[\left|\overline{\mathcal{T}^s}\right|\right] \le 1/q,$$

where the last inequality is because with probability $q$, $I_t = 1$. By combining the bounds for $t \in \overline{\mathcal{T}^s}$ and $t \in \underline{\mathcal{T}^s}$ together, lemma follows. $\qquad\square$

**Lemma D.5.** *Let $Q = \sum_{s \ge o} q_s$. Then,*

$$\mathbb{E}\left[\sum_{t\in\mathcal{T}^s} \min\left\{1, \|A_t^o\|_{V_{t-1}^{-1}}^2\right\}\right] \le 2d/Q + 1/Q.$$

*Proof.* Let $\det(V_{t-1}) = (q_{t-1})^d$. Let $x_t$ satisfies $\det(V_{t-1} + A_t^o(A_t^o)^\top) = (q_{t-1} + x_t)^d$. We let $I_t = \mathbb{1}\{I_t \ge o\}$. We divide $\mathcal{T}^s$ into two disjoint parts $\underline{\mathcal{T}^s}$ and $\overline{\mathcal{T}^s}$. Specifically,

- for $\underline{\mathcal{T}^s}$, it holds that for $t \in \underline{\mathcal{T}^s}$, $\det(V_{t-1} + A_t^o(A_t^o)^\top) \le 2^{d(s+1)}$.

- for $\overline{\mathcal{T}^s}$, it holds that for $t \in \overline{\mathcal{T}^s}$, $\det(V_{t-1} + A_t^o(A_t^o)^\top) > 2^{d(s+1)}$.

Note that for $I_t \geq o$,

$$\log(\det(V_t)) - \log(\det(V_{t-1} + A_t^o (A_t^o)^\top))$$

$$= \log(\det(V_t)) - \log(V_{t-1}) - \left( \log(\det(V_{t-1} + A_t^o (A_t^o)^\top)) - \log(V_{t-1}) \right)$$

$$\text{(due to Lemma D.1)}$$

$$= \log(1 + \|A_t^{I_t}\|_{V_{t-1}^{-1}}) - \log(1 + \|A_t^o\|_{V_{t-1}^{-1}})$$

$$\geq 0. \qquad\qquad\qquad\qquad\qquad\qquad\qquad\qquad\qquad\qquad\qquad\qquad \text{(due to Lemma D.3)}$$

Therefore, if we let $\det(V_{t-1} + A_t^{I_t} (A_t^{I_t})^\top) = (q_{t-1} + x_t')^d$ and $I_t \geq o$, then $x_t' \geq x_t$. Hence,

$$\sum_{t \in \underline{\mathcal{T}}^s} x_t \cdot I_t \leq 2^s.$$

Note that $\mathbb{P}(I_t \geq o) = \sum_{s \geq o} q_s = Q$. Let $\{\mathcal{F}_t\}_{t \in [T]}$ be the natural filtration of $\{A_t, I_t, X_t\}_{t \in [T]}$. We have

$$2^s \geq \mathbb{E}\left[\sum_{t \in \underline{\mathcal{T}}^s} x_t \cdot I_t\right]$$

$$= \mathbb{E}\left[\sum_{t \in [T]} x_t \cdot I_t \cdot \mathbb{1}\{t \in \underline{\mathcal{T}}^s\}\right]$$

$$= \sum_{t \in [T]} \mathbb{E}\left[x_t \cdot I_t \cdot \mathbb{1}\{t \in \underline{\mathcal{T}}^s\}\right]$$

$$= \sum_{t \in [T]} \mathbb{E}\left[\mathbb{E}\left[x_t \cdot I_t \cdot \mathbb{1}\{t \in \underline{\mathcal{T}}^s\} | \mathcal{F}_{t-1}\right]\right] \qquad\qquad \text{(Law of total expectation)}$$

$$= \sum_{t \in [T]} \mathbb{E}\left[x_t \cdot \mathbb{1}\{t \in \underline{\mathcal{T}}^s\} \cdot \mathbb{E}[I_t | \mathcal{F}_{t-1}]\right] \qquad (x_t \text{ and } \mathbb{1}\{t \in \underline{\mathcal{T}}^s\} \text{ are } \mathcal{F}_{t-1}\text{-measurable})$$

$$= \sum_{t \in [T]} \mathbb{E}\left[x_t \cdot \mathbb{1}\{t \in \underline{\mathcal{T}}^s\} \cdot Q\right] \qquad\qquad (I_t \text{ is an independent coin-tossing})$$

$$= Q\mathbb{E}\left[\sum_{t \in \underline{\mathcal{T}}^s} x_t\right]$$

and therefore

$$\mathbb{E}\left[\sum_{t \in \underline{\mathcal{T}}^s} x_t\right] \leq 2^s/Q.$$

For $t \in \underline{\mathcal{T}}^s$, we further have

$$\min\left\{1, \|A_t^o\|_{V_{t-1}^{-1}}^2\right\} \leq \min\left\{1, \|A_t^{I_t}\|_{V_{t-1}^{-1}}^2\right\}$$

$$\leq 2\log\left(\frac{\det(V_{t-1} + A_t^{I_t}(A_t^{I_t})^\top)}{\det(V_{t-1})}\right) \qquad\qquad \text{(due to Lemma D.1)}$$

$$= 2d\log\left(1 + \frac{x_t}{q_{t-1}}\right)$$

$$\leq 2d\log\left(1 + \frac{x_t}{2^s}\right) \qquad\qquad\qquad\qquad (\text{since } t \in \mathcal{T}^s, q_{t-1} \geq 2^s)$$

$$\leq \frac{2d \cdot x_t}{2^s} \qquad\qquad\qquad\qquad\qquad (\text{due to } \log(1 + x) \leq x \text{ for } x > 0.)$$

Therefore,

$$\mathbb{E}\left[\sum_{t\in\underline{\mathcal{T}}^s}\min\left\{1,\|A_t^o\|_{V_{t-1}^{-1}}^2\right\}\right]\leq\mathbb{E}\left[\sum_{t\in\underline{\mathcal{T}}^s}\frac{2d\cdot x_t}{2^s}\right]=\frac{2d}{2^s}\mathbb{E}\left[\sum_{t\in\underline{\mathcal{T}}^s}x_t\right]\leq 2d/Q.$$

From definition of $\overline{\mathcal{T}}^s$, if $I_t=1$ and $t\in\overline{\mathcal{T}}^s$, then $I_t\geq o$ and

$$\det(V_t)=\det(V_{n-1}+A_t^{I_t}(A_t^{I_t})^\top)>\det(V_{n-1}+A_t^o(A_t^o)^\top)>(s+1)^d.$$

Then, for all $\tau>t,\tau\notin\mathcal{T}^s$. Therefore, there is at most one $t\in\overline{\mathcal{T}}^s$ with $I_t=1$. We obtain

$$\mathbb{E}\left[\sum_{t\in\overline{\mathcal{T}}^s}\min\left\{1,\|A_t^o\|_{V_{t-1}^{-1}}^2\right\}\right]\leq\mathbb{E}\left[\left|\overline{\mathcal{T}}^s\right|\right]\leq 1/Q,$$

where the last inequality is because with probability $\sum_{s\geq o}q_s=Q$, $I_t=1$. By combining the bounds for $t\in\overline{\mathcal{T}}^s$ and $t\in\underline{\mathcal{T}}^s$ together, lemma follows. $\qquad\square$

# E   Experimental details

The code used in the experiments can be found in the following repository: `https://github.com/jajajang/sparsity_agnostic_model_selection`.

## E.1   Settings common to all algorithms

- Arm set: $\mathcal{A}_t=\mathcal{A}$ for all $t\in[T]$. $\mathcal{A}\subset\mathbb{S}^{d-1}$ (the unit sphere in $\mathbb{R}^d$) is a set of $d$-dimensional vectors drawn independently and uniformly at random from $\mathbb{S}^{d-1}$, with $d=16$ and $|\mathcal{A}|=30$.

- $\theta_*$ is an $S$-sparse ($S=1,2,4,8,16$) vector generated as follows: before the game starts, draw $(\theta_*)_1,\ldots,(\theta_*)_S\sim\mathbb{S}^{S-1}$, and $(\theta_*)_k=0$ for all $k>S$.

- The noise on rewards: $\{\varepsilon_t\}_{t\in[T]}$ are i.i.d. with $\xi_t\sim\text{Unif}([-1,1])$.

- Number of iterations: $T=10^4$.

- Number of models: $n=6$.

- Radius of confidence sets: $\alpha_0=0$, and $\alpha_i=2^i\log t$ for $i=1,\cdots,5$.

- Prior distribution $\{q_s\}_{s\in[6]}$

  - For `_Unif`, $\{q_s\}_{s\in[6]}=\left(\frac{1}{6},\frac{1}{6},\frac{1}{6},\frac{1}{6},\frac{1}{6},\frac{1}{6}\right)$
  - For `_Theory`, $\{q_s\}_{s\in[6]}=\left(\frac{C}{2},\frac{C}{4},\frac{C}{8},\frac{C}{16},\frac{C}{32},\frac{C}{64}\right)$ where $C=\frac{63}{64}$ is a normalizing constant.

- Each plot is the result of 20 repetitions for each method. The shade represents the 1-standard deviation bound.

- Hardware: Lenovo Thinkpad P16s Gen 2 Laptop - Type 21HL

  - CPU: 13th Gen Intel(R) Core(TM) i7-1360P 2.20 GHz
  - RAM: 32GB

- Computation time: total 1338.38 seconds.

## E.2   `AdaLinUCB` details

- Since we empirically observed that `Exp3` provided enough exploration, we aggressively set the forced exploration parameter $q$ to zero.

- The learning rate of `Exp3` was set to $\eta_t=2\sqrt{\frac{\log n}{nt}}$, see [5].

- Given the prior distribution $\{q_s\}_{s\in[6]}$, we set $P_t$ as follows:

$$P_{t,s}=\frac{q_s\exp\left(\eta_t S_{t,s}\right)}{\sum_{j=1}^n q_j\exp\left(\eta_t S_{t,j}\right)}$$

### E.3 OFUL **details**

- We used the log-determinant form of the confidence set based on Abbasi-Yadkori et al. [1, Theorem 2], which gives the choice

$$\sqrt{\gamma_t} = \sqrt{2 \log T + \log \det(V_t)} + 1$$

for the parameter $\gamma_t$ in (1.1) when $\lambda = 1$ and $\delta = 1/T$.

