# OpenReview forum: "Sparsity-Agnostic Linear Bandits with Adaptive Adversaries"
_NeurIPS.cc/2024/Conference — NeurIPS 2024 poster_

### Official Review · Reviewer_9XZb · 2024-06-24

**Soundness:** 3
**Presentation:** 3
**Contribution:** 3
**Rating:** 6
**Confidence:** 4

**Summary:**

This paper studies the sparse linear bandit problem without the prior knowledge of sparsity. The studied problem also considers general setup in which the context is chosen by an adaptive adversary and action set is not imposed with additional assumptions. Then, A OFUL based algorithms are proposed and regret bounds are provided.

**Strengths:**

- Lifting assumptions used in existing works while still maintaining comparable or even better regret bounds.

- The proposed algorithm provides an instance-dependent regret bound and worst-case bound as well.

- As the performance of SparseLinUCB highly depends on distribution q, different choices of q are shown.

- Numerical experiments are provided to support the significance of AdaLinUCB algorithm.

**Weaknesses:**

- The instance dependent regret bound shows no improvement compared with standard OFUL algorithm, and could be even worse than that of OFUL, e.g., S^2>d.

- For known sparsity and adaptive adversary setup, the instance depedent regret bound in Corollary 3.5 is worse than that of [2], which is dS/Delta.

- The regret bound of SparseLinUCB is sensitive to the choice of distribution q. Though authors provide a specific example on the choice of q, the regret bound now highly depends on the choice of constant C. For example, if one chooses C=1, the instance dependent bound is worse than that of OFUL when S^2>d.

- For the action chosen (step 5 in Algorithm 1), it could be difficult to compute A_t since this optimization problem contains non-linear term and the action set is arbitrary.

**Questions:**

Can authors provide another distribution selection case which can recover dS/Delta instance-dependent bound in known sparsity and adaptive adversary setup?

**Limitations:**

Yes

---

> ### Author Rebuttal · Authors · 2024-08-06
>
> We appreciate your valuable time and effort in offering detailed feedback on our work. In the following, we address your questions one by one.
>
> ---
>
> Q1: Can authors provide another distribution selection case which can recover $dS/\Delta$ instance-dependent bound in known sparsity and adaptive adversary setup?
>
> A1:  Yes, this can be achieved by setting $q_o = 1$. Then, Theorem 4 of [2] directly implies a problem-dependent bound of order $\tilde{O}\left(\frac{dS}{\Delta}\right)$ for $q_o = 1$. We will note this result in the revision.
>
> ----
>
> Q2: For known sparsity and adaptive adversary setup, the instance-dependent regret bound in Corollary 3.5 is worse than that of [2], which is dS/Delta.
>
> A2: See A1. For a known sparsity setup, our regret bound can be of the same order as that of [2].
>
> ---
>
> Q3: Why the instance dependent regret bound shows no improvement compared with standard OFUL algorithm.
>
> A3:  We have provided some intuitions on why it is difficult to provide a problem-dependent bound that benefits from sparsity in Lines 214-220. Essentially, as long as you set $q_{n}$ as a constant (even as small as $1/\sqrt{T}$), the regret will scale as $\tilde O(d^2 / \Delta)$.
>
> ---
>
> Q4: For the action chosen (step 5 in Algorithm 1), it could be difficult to compute $A_t$ since this optimization problem contains non-linear term and the action set is arbitrary.
>
> A4: The computation of $A_t$ in Step 5 of Algorithm 1 is as hard as computing the prediction in the OFUL algorithm, which can be done efficiently when the action set is finite. In the case of arbitrary actions sets, not much can be said about the computational efficiency of solving this problem, see [17, Section 19.3.1].

---

> > ### Comment · Reviewer_9XZb · 2024-08-11
> > **Official Comment by Reviewer 9XZb**
> >
> > Thanks for your response. I don't have further questions and will keep my score as is.

---

> > > ### Author Response · Authors · 2024-08-12
> > >
> > > Thank you for reviewing our response and for your support. If you have any further questions about our submission, please don't hesitate to reach out.

---

### Official Review · Reviewer_KeoW · 2024-06-28

**Soundness:** 3
**Presentation:** 3
**Contribution:** 3
**Rating:** 6
**Confidence:** 3

**Summary:**

This paper proposes statistically efficient linear bandit algorithms capable of handling cases where prior knowledge of the sparsity level $S$ is not given. The first algorithm, SparseLinUCB, achieves a $\tilde{O}(S \sqrt{dT})$ regret bound without any stochastic assumptions on the context vector, covering adversarially given context vectors. The main idea involves sampling the radius of the confidence set for the true reward parameter $\theta_*$ from a specific distribution and then selecting the optimistic action. It matches the lower bound when sparsity information is provided. The second algorithm, AdaLinUCB, updates the sampling distribution of the confidence radius using an approach (Exp3) that increases the likelihood of selecting a radius providing higher rewards. AdaLinUCB also achieves a $\tilde{O}(\sqrt{T})$ regret bound. Various experiments support the theoretical results of the proposed algorithms.

**Strengths:**

- The motivation for the problem addressed in the paper is well explained, and the related work is thoroughly described. Overall, the paper is well-written and easy to understand.
- The first algorithm, SparseLinUCB, is, to my knowledge, the first sparsity-agnostic linear bandit algorithm for adversarial context vectors. Additionally, it matches the lower bound regret when sparsity information is provided.
- The second algorithm, which updates the confidence radius distribution at each time step, is also very interesting. Previous works using similar methods achieved loose regret bounds ($\tilde{O}(T^{2/3})$), whereas the proposed algorithm achieves $\tilde{O}(\sqrt{T})$ regret (though I have not rigorously checked this proof).
- Various numerical experiments support the theory behind the proposed algorithms.

**Weaknesses:**

- The SparseLinUCB algorithm does not make stochastic assumptions about the action set (context vectors), thus providing theoretical guarantees even in the case of an adaptive adversary. However, AdaLinUCB is described as an algorithm for stochastic linear bandits. It seems that the stochastic assumptions required for AdaLinUCB's regret bound are not explained.
- The explanation about $n$ in the confidence radius distribution $\set{ q_s }_{s \in [n]}$ appears insufficient. Additionally, there seems to be no term for $ n $ in the regret bound. I am curious whether the regret bound is independent of $ n$.

**Questions:**

- (Related to the 1st bullet in Weaknesses) Can AdaLinUCB still achieve the currently presented regret bound if the action set (context vectors) is given by an adaptive adversary? If not, can you briefly explain the issue?
- (Related to the 2nd bullet in Weaknesses) It seems that how $\set{ q_s }_{s \in [n]}$ is determined would impact the regret bound. You introduced a specific distribution in Eq. (3.3). How does the regret bound change if $ n$ is significantly increased or decreased?
- The motivation for updating the confidence radius distribution at each time step in AdaLinUCB is interesting. However, the current result shows a looser regret bound compared to SparseLinUCB. Despite the computational cost of updating the distribution, there seems to be no statistical gain. In what instances would it be better to use AdaLinUCB over SparseLinUCB? Also, can you briefly explain why AdaLinUCB performs better empirically?

**Limitations:**

The authors have well-addressed the limitations and further research directions in Section 6. The content discussed in this paper appears to have little to no negative societal impact.

---

> ### Author Rebuttal · Authors · 2024-08-06
>
> We appreciate your valuable time and effort in offering detailed feedback on our work. In the following, we address your questions one by one.
>
> ----
>
> Q1: The SparseLinUCB algorithm does not make stochastic assumptions about the action set (context vectors), thus providing theoretical guarantees even in the case of an adaptive adversary. However, AdaLinUCB is described as an algorithm for stochastic linear bandits. It seems that the stochastic assumptions required for AdaLinUCB's regret bound are not explained.
>
> A1: AdaLinUCB is indeed designed to handle adaptive adversarial action sets (see, for example, the initial lines 414-419 of the proof where the adversarial nature of the action set appears in the analysis). We will clarify this in the revision. There may have been some confusion regarding the title of Section 4: "Adaptive Model Selection for Stochastic Linear Bandits."  In the literature, such as Chapter 19 of the bandit book [17] and reference [2], the term "stochastic linear bandits" typically refers to the presence of stochasticity in the subgaussian noise term $\epsilon_t$, as outlined in Equation (2.1).
>
> ---
>
> Q2: The explanation about $n$ in the confidence radius distribution $\\{q_s\\}_{s\in [n]}$ appears insufficient. Additionally, there seems to be no term for $n$ in the regret bound. I am curious whether the regret bound is independent of $n$.
>
> A2: As stated in Line 182, we choose $n = \Theta(\log d)$, which is large enough to ensure $\alpha_n \geq \gamma(1/T)$. We will clarify this fact in the revision.
>
> ---
>
> Q3: (Related to the 2nd bullet in weaknesses) It seems that how $\\{q_s\\}_{s\in [n]}$ is determined would impact the regret bound. You introduced a specific distribution in Eq. (3.3). How does the regret bound change if $n$ is significantly increased or decreased.
>
> A3: Increasing $n$ beyond $\Theta(\log d)$ significantly affects the regret bound of Theorem 3.2 through the choice of $q_s$. If we increase $n$, the regret upper bound approximately becomes $O(n\sqrt{dT})$. Indeed, following our choice $q_i \approx 2^{−i}$ (Eq. (3.3)), the first term in Lines 190-191 has order $O(n\sqrt{dT})$, while the second term has order $O(S \sqrt{dT})$ since $Q\approx \sum_{i>o} 2^{-i} \approx 2^{-o}\approx O(1/S)$, regardless of the size of $n$. Therefore, the final regret bound becomes of order $O(n\sqrt{dT})$. Intuitively, this makes sense, as increasing the number of models increases the exploration, which eventually has a negative impact on the regret.
>
> If $n$ is decreased such that $\alpha_n < \gamma(1/T)$, then $\theta^*$ may not belong to any confidence set in our hierarchy, and the regret could thus become linear in $T$. As such, we must ensure $\alpha_{n}\geq \gamma(1/T)$ and highlight it in Line 181.
>
> ---
>
> Q4: The motivation for updating the confidence radius distribution at each time step in AdaLinUCB is interesting. However, the current result shows a looser regret bound compared to SparseLinUCB. Despite the computational cost of updating the distribution, there seems to be no statistical gain. In what instances would it be better to use AdaLinUCB over SparseLinUCB? Also, can you briefly explain why AdaLinUCB performs better empirically?
>
> A4: AdaLinUCB works better empirically since it automatically adjusts the radius of the confidence bound. Theoretical confidence bounds are typically set very conservatively to ensure error probability guarantees for hard environments (e.g., when the action set is chosen by the adaptive adversary), which often results in over-exploration. For example, in Chapter 36 of the Szepesvari and Lattimore book "Bandits," the sixth note in Section 36.5 mentions that values of the Linear Thompson Sampling parameter (i.e., the variance of the posterior distribution playing a similar role to the radius of the confidence bound in UCB) which show good empirical performance do not have any theoretical guarantee. Conversely, the values of the parameter ensuring a solid theoretical guarantee consistently perform poorly in practice.
>
> As a future research direction, we can further explore the gap between the practical superior performance and theoretical limitations of AdaLinUCB, and even consider what the lower bound for sparsity-agnostic linear bandits might be.

---

> > ### Comment · Reviewer_KeoW · 2024-08-12
> >
> > Thank you for the detailed explanation. I have no further questions. I have kept my rating, as my original score was already positive and supportive of accepting the paper!

---

> > > ### Author Response · Authors · 2024-08-12
> > >
> > > Thank you for reviewing our response and for your support. If you have any further questions regarding our submission, please feel free to reach out.

---

### Official Review · Reviewer_muo1 · 2024-07-01

**Soundness:** 3
**Presentation:** 3
**Contribution:** 3
**Rating:** 5
**Confidence:** 3

**Summary:**

This paper studies the stochastic linear bandits when the action set can be arbitrarily chosen without some additional assumptions. And the authors propose a randomized sparsity-agnostic bandit algorithm using the model selection idea, and show that EXP3 can be used as the master algorithm to obtain a decent regret bound. Experimental results are included in the end to verify the high efficiency of the proposed algorithms.

**Strengths:**

1. The paper is easy to read, and most parts are pretty clear. E.g. Table 1 helps reader catch up all the existing literature and their differences quickly.
2. This paper studies an interesting and important problem when the arm set is arbitrarily chosen under the sparse linear bandit problem. I didn't check the proof in appendix in detail but all arguments seem reasonable to me.
3. Empirical results are provided to illustrate the high efficiency of the proposed algorithms.

**Weaknesses:**

1. The used techniques are based on the existing literature (e.g. seqsew, exp3 master algorithm). It will be better to show the theory novelty of this work in a separate paragraph.

2. Lower bounds are not provided, as the authors mention in the limitation parts. So it is a little bit hard for readers to justify how good are the proposed bounds.

**Questions:**

1. Why is the boundness of the random noise necessary in your theoretical proof?
2. Is it possible to bound the regret with high probability instead of the expected value? If some existing literature proposed the regret bound with high probability, then it may not be fair to do the direct order comparison in Table 1.
3. Can you report the running time of your method in the experiments?

**Limitations:**

No negative societal impact.

---

> ### Author Rebuttal · Authors · 2024-08-06
>
> We appreciate your valuable time and effort in offering detailed feedback on our work. In the following, we address your questions one by one.
>
> ---
>
> Q1: The used techniques are based on the existing literature (e.g. seqsew, exp3 master algorithm). It will be better to show the theory novelty of this work in a separate paragraph.
>
> A1: We do have a 40-line paragraph “Technical challenges” (Lines 61-100) explaining the theoretical novelty.  One of our main technical contributions is demonstrating that our exploration, measured by $\det V_{t-1}$, grows very rapidly, as highlighted in Lines 81-86 (the proof details are shown in Lemmas D.4 and D.5). As an example to show the power of our new techniques, we achieve a $\sqrt{T}$ regret bound for AdaLinUCB, whereas previous works [11,24] using the Exp3 to learn the probability over a hierarchy of confidence sets have a regret bound of only $T^{2/3}$.
>
> Please refer to the Technical challenges paragraph for a full list of our original contributions.
>
> ----
>
> Q2: Why is the boundness of the random noise necessary in your theoretical proof?
>
> A2: Note that our results in Section 3 do not require boundedness of the noise, but only 1-subgaussianity, see (2.1) in Lines 138-139. We only require $\epsilon_t \in [-1,1]$ in Section 4 because the Exp3 algorithm, which is used in the proof of Theorem 4.1, requires bounded rewards, which in turn requires the noise to be deterministically bounded.
>
> ---
>
> Q3: Is it possible to bound the regret with high probability instead of the expected value? If some existing literature proposed the regret bound with high probability, then it may not be fair to do the direct order comparison in Table 1.
>
> A3: Thanks for pointing out this interesting question. We believe our in-expectation regret bound for SparseLinUCB could be extended to a bound in high probability after some changes in the analysis. As these changes do not appear to be trivial as they require a somewhat intricate martingale concentration analysis, we will leave this extension to future work.
>
> In the revised version, we will also add notes to the table highlighting the algorithms that are high probability.
>
> ---
>
> Q4: Can you report the running time of your method in the experiments?
>
> A4: Although there were slight differences depending on the sparsity level or initial probability distribution, on average, a run of OFUL took 0.7 seconds, a run of SparseLinUCB took 1.4 seconds, and finally, a run of AdaLinUCB took 1.9 seconds. The longer runtime of SparseLinUCB and AdaLinUCB is motivated by the need of managing an ensemble of models. However, as we did not optimize the code, we expect these runtimes to be improvable.

---

> > ### Comment · Reviewer_muo1 · 2024-08-11
> >
> > Thank you for your responses. I have no more questions.

---

> > > ### Author Response · Authors · 2024-08-11
> > >
> > > Thank you for your feedback. If you have any further questions about our submission, please do not hesitate to reach out. We highly value your perspective, and should you find our responses satisfactory, we would be grateful if you would consider raising your score for our paper.

---

### Official Review · Reviewer_zJ6Y · 2024-07-09

**Soundness:** 3
**Presentation:** 1
**Contribution:** 2
**Rating:** 5
**Confidence:** 4

**Summary:**

This paper studies Linear bandits with adversaries when the underlying parameter $\theta$ is sparse. It combines a previous sparse linear regression algorithm named SeqSEW with LinUCB, proposing an algorithm named SparseLinUCB. It also proposes a variant of the EXP3 algorithm named AdaLinUCB. Regret bounds dependent on the sparsity dimension $S$ are proved. Experiments are conducted on synthetic data.

**Strengths:**

1. The paper proposes algorithms for sparse linear bandits with adversarial action sets. It proved a regret upper bound better than previous when the sparsity dimension $S$ is quite small with no assumptions on the sparsity structure. If the sparsity level is known, the result is optimal.
2. It also provides an instance-dependent regret bound.
3. It conducts synthetic experiments to show its better performance.

**Weaknesses:**

1. The paper misses many very related works, especially those on K-armed bandits and linear bandits.

2. For the main algorithm design, I cannot see the necessity to use an online learning oracle to predict the reward $\hat X_t$ and then calculate the least squares estimation when having access to the real reward $X_t$. Could you explain it further?

3. The writing is unclear.

(1) In Line 176,  "the distribution $\\{q_i\\} _ {i \in [n]}$" is not well defined, over what?

(2) The notation of the confidence set in Line 177 is contradictory with the notation in Appendix A, as it uses 2-norm rather than the $V_{t-1}$ norm.

(3) In equation 3.1, $\gamma$ is a problem-dependent term so you cannot set the exact value of $o$, as is done in Corollary 3.3.

(4) Line 240: in terms of what?

4. Line 249: “the parameter q does not provide a similar flexibility”, so I do not see the use of this probable selection, what is the benefit of the new algorithm design in Section 4?

**Questions:**

See Weaknesses

**Limitations:**

The authors have addressed the limitations and societal impact.

---

> ### Author Rebuttal · Authors · 2024-08-06
>
> Thank you for your valuable time and effort in providing detailed feedback on our work. We now address your questions one by one.
>
> ---
> Q1: The paper misses many very related works, especially those on K-armed bandits and linear bandits.
>
> A1:  It would be useful to have some concrete pointers to the missing works, but we will definitely browse more carefully through the relevant literature.
>
> ---
>
> Q2: For the main algorithm design, I cannot see the necessity to use an online learning oracle to predict the reward.
>
> A2:  The use of an online linear regression oracle for sparse linear bandits is an established technique whose advantages in the analysis have been shown in [2] and also in bandit book [17], see our comment in Line 149.  In particular, using a sparse online learning algorithm provides a dependence on the sparsity parameter $S$ in the confidence bound. We therefore use Lemmas 2.1 and 2.2 from [2] to set $\gamma(1/T) = O(S\log T)$. Thus, there exists some constant $C$ such that
>
> $\mathcal C_t=\\{ \theta \in \mathbb R^d: \|\theta\|_2^2+ \sum_s(\hat X_s - \langle A_s, \theta \rangle)^2 \leq CS\log T \\}$.
>
>
> It is unclear how such a dependence could be obtained using the standard least square estimator, which would only provide a dependence on $d$.
>
> ---
>
> Q3: Line 249: “the parameter $q$ does not provide a similar flexibility”, so I do not see the use of this probable selection, what is the benefit of the new algorithm design in Section 4?
>
> A3: The main advantage of AdaLinUCB is its empirical performance, as we explain in Lines 243-244 and demonstrate in our experiments where it outperforms both OFUL and SparseLinUCB. This improved empirical performance is due to AdaLinUCB’s ability of tuning its distribution based on observed rewards.
>
> From a theoretical viewpoint, AdaLinUCB achieves a regret bound of $\tilde{O}(\sqrt{T})$, using novel techniques descibed in Lines 61-100. Previous analyses of Exp3-based algorithms in [11] and [24] only achieved $\tilde{O}(T^{2/3})$ regret bounds, even under i.i.d. assumptions on the generation of the action sets, see Lines 122-131 for a description of these related works.
>
> ---
>
> Q4: In Line 176, “the distribution $\\{q_i\\}_{i\in[n]}$” is not well defined, over what?
>
> A4: In Lines 174-176 we say that the algorithm uses a hierarchy of confidence sets of increasing radius and draws the index $I_t$ of the confidence set by sampling from the probability distribution $\\{q_i\\}_{i \in [n]}$. We will clarify that the distribution is over the indices of the confidence sets, and make it explicit that $n$ is an input parameter to SparseLinUCB.
>
> ---
>
> Q5: The notation of the confidence set in 177.
>
> A5:  This is a typo. It should be $V_{t-1}$ norm. Thank you for your careful reading!
>
> ---
>
> Q6: In equation 3.1, $\gamma$ is a problem-dependent term so you cannot set the exact value of $o$, as is done in Corollary 3.3.
>
> A6: In Corollary 3.3, we explicitly state that it is assumed the sparsity level $S$ is known, as detailed in Lines 194-195. With $S$ known, the upper bound of $\gamma(\delta)$ can be easily calculated from its definition.
>
> ---
> Q7: Line 240: in terms of?
>
> A7: This is a typo. We will remove the words “in terms of”.

---

> > ### Comment · Reviewer_zJ6Y · 2024-08-11
> >
> > Thanks for your detailed responses, which do solve most of my concerns. I have increased my scores to reflect this. However, I suggest polishing the writing to help with understanding. Good luck!

---

> > > ### Author Response · Authors · 2024-08-12
> > >
> > > Thank you for reviewing our response and for the improved score. We will incorporate your valuable suggestions and make the necessary changes in the revised version of the paper. If you have any further questions about our submission, please don't hesitate to reach out.

---

### Decision · Program_Chairs · 2024-09-25

**Decision:**

Accept (poster)

**Comment:**

This paper studies sparse linear contextual bandits, and developed new regret bounds when the sparsity is unknown. The reviews agree that the paper makes descent contribution to the problem.